# On Top-k Selection in Multi-Armed Bandits and Hidden Bipartite Graphs

Wei Cao[1]          Jian Li[1]          Yufei Tao[2]          Zhize Li[1]

[1]Tsinghua University    [2]Chinese University of Hong Kong

[1]{cao-w13@mails, lijian83@mail, zz-li14@mails}.tsinghua.edu.cn    [2]taoyf@cse.cuhk.edu.hk

## Abstract

This paper discusses how to efficiently choose from $n$ unknown distributions the $k$ ones whose means are the greatest by a certain metric, up to a small relative error. We study the topic under two standard settings—*multi-armed bandits* and *hidden bipartite graphs*—which differ in the nature of the input distributions. In the former setting, each distribution can be sampled (in the i.i.d. manner) an arbitrary number of times, whereas in the latter, each distribution is defined on a population of a finite size $m$ (and hence, is fully revealed after $m$ samples). For both settings, we prove lower bounds on the total number of samples needed, and propose optimal algorithms whose sample complexities match those lower bounds.

## 1   Introduction

This paper studies a class of problems that share a common high-level objective: from a number $n$ of probabilistic distributions, find the $k$ ones whose means are the greatest by a certain metric.

**Crowdsourcing.** A crowdsourcing algorithm (see recent works [1, 13] and the references therein) summons a certain number, say $k$, of individuals, called *workers*, to collaboratively accomplish a complex task. Typically, the algorithm breaks the task into a potentially very large number of micro-tasks, each of which makes a binary decision (*yes* or *no*) by taking the majority vote from the participating workers. Each worker is given an (often monetary) reward for every micro-task that s/he participates in. It is therefore crucial to identify the most reliable workers that have the highest rates of making correct decisions. Because of this, a crowdsourcing algorithm should ideally be preceded by an *exploration phase*, which selects the best $k$ workers from $n$ candidates by a series of "control questions". Every control-question must be paid for in the same way as a micro-task. The challenge is to find the best workers with the least amount of money.

**Frequent Pattern Discovery.** Let $B$ and $W$ be two relations. Given a join predicate $Q(b, w)$, the *joining power* of a tuple $b \in B$ equals the number of tuples $w \in W$ such that $b$ and $w$ satisfy $Q$. A *top-k semi-join* [14, 17] returns the $k$ tuples in $B$ with the greatest joining power. This type of semi-joins is notoriously difficult to process when the evaluation of $Q$ is complicated, and thus unfriendly to tailored-made optimization. A well-known example from graph databases is the discovery of frequent patterns [14], where $B$ is a set of graph patterns, $W$ a set of data graphs, and $Q(b, w)$ decides if a pattern $b$ is a subgraph of a data graph $w$. In this case, top-$k$ semi-join essentially returns the set of $k$ graph patterns most frequently found in the data graphs. Given a black box for resolving subgraph isomorphism $Q(b, w)$, the challenge is to minimize the number of calls to the black box. We refer to the reader to [14, 15] for more examples of difficult top-$k$ semi-joins of this sort.

### 1.1   Problem Formulation

The paper studies four problems that capture the essence of the above applications.

**Multi-Armed Bandit.** We consider a standard setting of stochastic multi-armed bandit selection. Specifically, there is a bandit with a set $B$ of $n$ arms, where the $i$-th arm is associated with a Bernoulli

distribution with an unknown mean $\theta_i \in (0, 1]$. In each round, we choose an arm, pull it, and then collect a reward, which is an i.i.d. sample from the arm's reward distribution.

Given a subset $V \subseteq B$ of arms, we denote by $a_i(V)$ the arm with the $i$-th largest mean in $V$, and by $\theta_i(V)$ the mean of $a_i(V)$. Define $\theta_{\mathsf{avg}}(V) = \frac{1}{k} \sum_{i=1}^{k} \theta_i(V)$, namely, the average of the means of the top-$k$ arms in $V$.

Our first two problems aim to identify $k$ arms whose means are the greatest either individually or aggregatively:

**Problem 1** [Top-$k$ Arm Selection ($k$-AS)] *Given parameters $\epsilon \in \left(0, \frac{1}{4}\right), \delta \in \left(0, \frac{1}{48}\right)$, and $k \leq n/2$, we want to select a $k$-sized subset $V$ of $B$ such that, with probability at least $1 - \delta$, it holds that*

$$\theta_i(V) \geq (1 - \epsilon)\theta_i(B), \ \forall i \leq k.$$

We further study a variation of $k$-AS where we change the multiplicative guarantee $\theta_i(V) \geq (1 - \epsilon)\theta_i(B)$ to an additive guarantee $\theta_i(V) \geq \theta_i(B) - \epsilon'$. We refer to the modified problem as Top-$k_{\mathsf{add}}$ Arm Selection($k_{\mathsf{add}}$-AS). Due to the space constraint, we present all the details of $k_{\mathsf{add}}$-AS in Appendix C.

**Problem 2** [Top-$k_{\mathsf{avg}}$ Arm Selection ($k_{\mathsf{avg}}$-AS)] *Given the same parameters as in $k$-AS, we want to select a $k$-sized subset $V$ of $B$ such that, with probability at least $1 - \delta$, it holds that*

$$\theta_{\mathsf{avg}}(V) \geq (1 - \epsilon)\theta_{\mathsf{avg}}(B).$$

For both problems, the *cost* of an algorithm is the total number of arms pulled, or equivalently, the total number of samples drawn from the arms' distributions. For this reason, we refer to the cost as the algorithm's *sample complexity*. It is easy to see that $k$-AS is more stringent than $k_{\mathsf{avg}}$-AS; hence, a feasible solution to the former is also a feasible solution to the latter, but not the vice versa.

**Hidden Bipartite Graph.** The second main focus of the paper is the exploration of *hidden bipartite graphs*. Let $G = (B, W, E)$ be a bipartite graph, where the nodes in $B$ are colored black, and those in $W$ colored white. Set $n = |B|$ and $m = |W|$. The edge set $E$ is *hidden* in the sense that an algorithm does not see any edge at the beginning. To find out whether an edge exists between a black vertex $b$ and a white vertex $w$, the algorithm must perform a *probe* operation. The *cost* of the algorithm equals the number of such operations performed.

If an edge exists between $b$ and $w$, we say that there is a *solid edge* between them; otherwise, we say that they have an *empty edge*. Let $\deg(b)$ be the degree of a black vertex $b$, namely, the number of solid edges of $b$. Given a subset of black vertices $V \subseteq B$, we denote by $b_i(V)$ the black vertex with $i$-th largest degree in $V$, and by $\deg_i(V)$ the degree of $b_i(V)$. Furthermore, define $\deg_{\mathsf{avg}}(V) = \frac{1}{k} \sum_{i=1}^{k} \deg_i(V)$.

We now state the other two problems studied in this work, which aim to identify $k$ black vertices whose degrees are the greatest either individually or aggregatively:

**Problem 3** [$k$-Most Connected Vertex [14] ($k$-MCV)] *Given parameters $\epsilon \in \left(0, \frac{1}{4}\right), \delta \in \left(0, \frac{1}{48}\right)$, and $k \leq n/2$, we want to select a $k$-sized subset $V$ of $B$ such that, with probability at least $1 - \delta$, it holds that*

$$\deg_i(V) \geq (1 - \epsilon)\deg_i(B), \ \forall i \leq k.$$

**Problem 4** [$k_{avg}$-Most Connected Vertex ($k_{\mathsf{avg}}$-MCV)] *Given the same parameters as in $k$-MCV, we want to select a $k$-sized subset $V$ of $B$ such that, with probability at least $1 - \delta$, it holds that*

$$\deg_{\mathsf{avg}}(V) \geq (1 - \epsilon)\deg_{\mathsf{avg}}(B).$$

A feasible solution to $k$-MCV is also feasible for $k_{\mathsf{avg}}$-MCV, but not the vice versa. We will refer to the cost of an algorithm also as its *sample complexity*, by regarding a probe operation as "sampling" the edge probed. For any deterministic algorithm, the adversary can force the algorithm to always probe $\Omega(mn)$ edges. Hence, we only consider randomized algorithms.

$k$-MCV can be reduced to $k$-AS. Given a hidden bipartite graph $(B, W, E)$, we can treat every black vertex $b \in B$ as an "arm" associated with a Bernoulli reward distribution: the reward is 1 with probability $\deg(b)/m$ (recall $m = |W|$), and 0 with probability $1 - \deg(b)/m$. Any algorithm $\mathcal{A}$ for $k$-AS can be deployed to solve $k$-MCV as follows. Whenever $\mathcal{A}$ samples from arm $b$, we randomly choose a white vertex $w \in W$, and probe the edge between $b$ and $w$. A reward of 1 is returned to $\mathcal{A}$ if and only if the edge exists.

$k$-AS and $k$-MCV differ, however, in the size of the population that a reward distribution is defined on. For $k$-AS, the reward of each arm is sampled from a population of an *indefinite* size, which can even be infinite. Consequently, $k$-AS nicely models situations such as the crowdsourcing application mentioned earlier.

For $k$-MCV, the reward distribution of each "arm" (i.e., a black vertex $b$) is defined on a population of size $m = |W|$ (i.e., the edges of $b$). This has three implications. First, $k$-MCV is a better modeling of applications like top-$k$ semi-join (where an edge exists between $b \in B$ and $w \in W$ if and only if $Q(b, w)$ is true). Second, the problem admits an obvious algorithm with cost $O(nm)$ (recall $n = |B|$): simply probe all the hidden edges. Third, an algorithm never needs to probe the same edge between $b$ and $w$ twice—once probed, whether the edge is solid or empty is perpetually revealed. We refer to the last implication as the *history-awareness property*.

The above discussion on $k$-AS and $k$-MCV also applies to $k_{\mathsf{avg}}$-AS and $k_{\mathsf{avg}}$-MCV. For each of above problems, we refer to an algorithm which achieves the precision and failure requirements prescribed by $\epsilon$ and $\delta$ as an $(\epsilon, \delta)$-*approximate* algorithm.

## 1.2 Previous Results

**Problem 1.** Sheng et al. [14] presented an algorithm[1] that solves $k$-AS with expected cost $O(\frac{n}{\epsilon^2} \frac{1}{\theta_k(B)} \log \frac{n}{\delta})$. No lower bound is known on the sample complexity of $k$-AS. The closest work is due to Kalyanakrishnan et al. [11]. They considered the EXPLORE-$k$ problem, where the goal is to return a set $V$ of $k$ arms such that, with probability at least $1 - \delta$, the mean of each arm in $V$ is at least $\theta_k(B) - \epsilon'$. They showed an algorithm with sample complexity $\Theta(\frac{n}{\epsilon'^2} \log \frac{k}{\delta})$ in expectation and establish a matching lower bound. Note that EXPLORE-$k$ ensures an *absolute-error* guarantee, which is weaker than the *individually relative-error* guarantee of $k$-AS. Therefore, the same EXPLORE-$k$ lower bound also applies to $k$-AS.

The readers may be tempted to set $\epsilon' = \epsilon \cdot \theta_k(B)$ to derive a "lower bound" of $\Omega(\frac{n}{\epsilon^2} \frac{1}{(\theta_k(B))^2} \log \frac{k}{\delta})$ for $k$-AS. This, however, is clearly wrong because when $\theta_k(B) = o(1)$ (a typical case in practice) this "lower bound" may be even higher than the upper bound of [14] mentioned earlier. The cause of the error lies in that the hard instance constructed in [11] requires $\theta_k(B) = \Omega(1)$.

**Problem 2.** The $O(\frac{n}{\epsilon^2} \frac{1}{\theta_k(B)} \log \frac{n}{\delta})$ upper bound of [14] on $k$-AS carries over to $k_{\mathsf{avg}}$-AS (which, as mentioned before, can be solved by any $k$-AS algorithm). Zhou et al. [16] considered an OPTMAI problem whose goal is to find a $k$-sized subset $V$ such that $\theta_{\mathsf{avg}}(V) - \theta_{\mathsf{avg}}(B) \leq \epsilon'$ holds with probability at least $1 - \delta$. Note, once again, that this is an absolute-error guarantee, as opposed to the relative-error guarantee of $k_{\mathsf{avg}}$-AS. For OPTMAI, Zhou et al. presented an algorithm with sample complexity $O(\frac{n}{\epsilon'^2}(1 + \frac{\log(1/\delta)}{k}))$ in expectation. Observe that if $\theta_{\mathsf{avg}}(B)$ is available *magically* in advance, we can immediately apply the OPTMAI algorithm of [16] to settle $k_{\mathsf{avg}}$-AS by setting $\epsilon' = \epsilon \cdot \theta_{\mathsf{avg}}(B)$. The expected cost of the algorithm becomes $O(\frac{n}{\epsilon^2} \frac{1}{(\theta_{\mathsf{avg}}(B))^2}(1 + \frac{\log(1/\delta)}{k}))$ (which is suboptimal. See the table).

No lower bound is known on the sample complexity of $k_{\mathsf{avg}}$-AS. For OPTMAI, Zhou et al. [16] proved a lower bound of $\Omega(\frac{n}{\epsilon'^2}(1 + \frac{\log(1/\delta)}{k}))$, which directly applies to $k_{\mathsf{avg}}$-AS due to its stronger quality guarantee.

**Problems 3 and 4.** Both problems can be trivially solved with cost $O(nm)$. Furthermore, as explained in Section 1.1, $k$-MCV and $k_{\mathsf{avg}}$-MCV can be reduced to $k$-AS and $k_{\mathsf{avg}}$-AS respectively. Indeed, the best existing $k$-AS and $k_{\mathsf{avg}}$-AS algorithms (surveyed in the above) serve as the state of the art for $k$-MCV and $k_{\mathsf{avg}}$-MCV, respectively.

Prior to this work, no lower bound results were known for $k$-MCV and $k_{\mathsf{avg}}$-MCV. Note that *none* of the lower bounds for $k$-AS (or $k_{\mathsf{avg}}$-AS) is applicable to $k$-MCV (or $k_{\mathsf{avg}}$-MCV, resp.), because there is no reduction from the former problem to the latter.

## 1.3 Our Results
We obtain tight upper and lower bounds for all of the problems defined in Section 1.1. Our main results are summarized in Table 1 (all bounds are in expectation). Next, we explain several highlights, and provide an overview into our techniques.

Table 1: Comparison of our and previous results (all bounds are in expectation)

| problem | | sample complexity | source |
|---|---|---|---|
| $k$-AS | upper bound | $O\left(\frac{n}{\epsilon^2}\frac{1}{\theta_k(B)}\log\frac{n}{\delta}\right)$ | [14] |
| | | $O\left(\frac{n}{\epsilon^2}\frac{1}{\theta_k(B)}\log\frac{k}{\delta}\right)$ | **new** |
| | lower bound | $\Omega\left(\frac{n}{\epsilon^2}\log\frac{k}{\delta}\right)$ | [11] |
| | | $\Omega\left(\frac{n}{\epsilon^2}\frac{1}{\theta_k(B)}\log\frac{k}{\delta}\right)$ | **new** |
| $k_{\mathsf{avg}}$-AS | upper bound | $O(\frac{n}{\epsilon^2}\frac{1}{\theta_k(B)}\log\frac{n}{\delta})$ | [14] |
| | | $O\left(\frac{n}{\epsilon^2}\frac{1}{(\theta_{\mathsf{avg}}(B))^2}\left(1+\frac{\log(1/\delta)}{k}\right)\right)$ | [16] |
| | | $O\left(\frac{n}{\epsilon^2}\frac{1}{\theta_{\mathsf{avg}}(B)}\left(1+\frac{\log(1/\delta)}{k}\right)\right)$ | **new** |
| | lower bound | $\Omega\left(\frac{n}{\epsilon^2}\left(1+\frac{\log(1/\delta)}{k}\right)\right)$ | [16] |
| | | $\Omega\left(\frac{n}{\epsilon^2}\frac{1}{\theta_{\mathsf{avg}}(B)}\left(1+\frac{\log(1/\delta)}{k}\right)\right)$ | **new** |
| $k$-MCV | upper bound | $O\left(\min\left\{\frac{n}{\epsilon^2}\frac{m}{\deg_k(B)}\log\frac{n}{\delta},nm\right\}\right)$ | [14] |
| | | $O\left(\min\left\{\frac{n}{\epsilon^2}\frac{m}{\deg_k(B)}\log\frac{k}{\delta},nm\right\}\right)$ | **new** |
| | lower bound | $\begin{cases}\Omega\left(\frac{n}{\epsilon^2}\frac{m}{\deg_k(B)}\log\frac{k}{\delta}\right)\text{ if }\deg_k(B)\geq\Omega(\frac{1}{\epsilon^2}\log\frac{n}{\delta})\\ \Omega(nm)\text{ if }\deg_k(B)<O(\frac{1}{\epsilon})\end{cases}$ | **new** |
| $k_{\mathsf{avg}}$-MCV | upper bound | $O\left(\min\left\{\frac{n}{\epsilon^2}\frac{m}{\deg_k(B)}\log\frac{n}{\delta},nm\right\}\right)$ | [14] |
| | | $O\left(\min\left\{\frac{n}{\epsilon^2}\frac{m^2}{(\deg_{\mathsf{avg}}(B))^2}\left(1+\frac{\log(1/\delta)}{k}\right),nm\right\}\right)$ | [16] |
| | | $O\left(\min\left\{\frac{n}{\epsilon^2}\frac{m}{\deg_{\mathsf{avg}}(B)}\left(1+\frac{\log(1/\delta)}{k}\right),nm\right\}\right)$ | **new** |
| | lower bound | $\begin{cases}\Omega\left(\frac{n}{\epsilon^2}\frac{m}{\deg_{\mathsf{avg}}(B)}\left(1+\frac{\log(1/\delta)}{k}\right)\right)\\ \qquad\qquad\text{ if }\deg_{\mathsf{avg}}(B)\geq\Omega(\frac{1}{\epsilon^2}\log\frac{n}{\delta})\\ \Omega(nm)\text{ if }\deg_{\mathsf{avg}}(B)<O(\frac{1}{\epsilon})\end{cases}$ | **new** |

**$k$-AS.** Our algorithm improves the $\log n$ factor of [14] to $\log k$ (in practice $k\ll n$), thereby achieving the optimal sample complexity (Theorem 1).

Our analysis for $k$-AS is inspired by [8, 10, 11] (in particular the *median elimination* technique in [8]). However, the details are very different and more involved than the previous ones (the application of median elimination of [8] was in a much simpler context where the analysis was considerably easier). On the lower bound side, our argument is similar to that of [11], but we need to get rid of the $\theta_k(B)=\Omega(1)$ assumption (as explained in Section 1.2), which requires several changes in the analysis (Theorem 2).

**$k_{\mathsf{avg}}$-AS.** Our algorithm improves both existing solutions in [14, 16] significantly, noticing that both $\theta_k(B)$ and $(\theta_{\mathsf{avg}}(B))^2$ are never larger, but can be far smaller, than $\theta_{\mathsf{avg}}(B)$. This improvement results from an enhanced version of median elimination, and once again, requires a non-trivial analysis specific to our context (Theorem 4). Our lower bound is established with a novel reduction from the 1-AS problem (Theorem 5). It is worth nothing that the reduction can be used to simplify the proof of the lower bound in [16, Theorem 5.5] .

**$k$-MCV and $k_{\mathsf{avg}}$-MCV.** The stated upper bounds for $k$-MCV and $k_{\mathsf{avg}}$-MCV in Table 1 can be obtained directly from our $k$-AS and $k_{\mathsf{avg}}$-AS algorithms. In contrast, all the lower-bound arguments for $k$-AS and $k_{\mathsf{avg}}$-AS—which crucially rely on the samples being i.i.d.—break down for the two MCV problems, due to the history-awareness property explained in Section 1.1.

For $k$-MCV, we remedy the issue by (i) (when $\deg_k(B)$ is large) a reduction from $k$-AS, and (ii) (when $\deg_k(B)$ is small) a reduction from a sampling lower bound for distinguishing two extremely similar distributions (Theorem 3). Analogous ideas are deployed for $k_{\mathsf{avg}}$-MCV (Theorem 6). Note that for a small range of $\deg_k(B)$ (i.e., $\Omega(\frac{1}{\epsilon})<\deg_k(B)<O(\frac{1}{\epsilon^2}\log\frac{n}{\delta})$), we do not have the optimal lower bounds yet for $k$-MCV and $k_{\mathsf{avg}}$-MCV. Closing the gap is left as an interesting open problem.

**Algorithm 1:** ME-AS

**1 input:** $B, \epsilon, \delta, k$
**2 for** $\mu = 1/2, 1/4, \ldots$ **do**
**3**      $S = \text{ME}(B, \epsilon, \delta, \mu, k)$;
**4**      $\{(a_i, \hat{\theta}^{US}(a_i)) \mid 1 \leq i \leq k\} = \text{US}(S, \epsilon, \delta, (1 - \epsilon/2)\mu, k)$;
**5**      **if** $\hat{\theta}^{US}(a_k) \geq 2\mu$ **then**
**6**          **return** $\{a_1, \ldots, a_k\}$;

---

**Algorithm 2:** Median Elimination (ME)

**1 input:** $B, \epsilon, \delta, \mu, k$
**2** $S_1 = B, \epsilon_1 = \epsilon/16, \delta_1 = \delta/8, \mu_1 = \mu$, and $\ell = 1$;
**3 while** $|S_\ell| > 4k$ **do**
**4**      sample every arm $a \in S_\ell$ for $Q_\ell = (12/\epsilon_\ell^2)(1/\mu_\ell) \log(6k/\delta_\ell)$ times;
**5**      **for** *each arm* $a \in S_\ell$ **do**
**6**          its empirical value $\hat{\theta}(a)$ = the average of the $Q_\ell$ samples from $a$;
**7**      $a_1, \ldots, a_{|S_\ell|}$ = the arms sorted in non-increasing order of their empirical values;
**8**      $S_{\ell+1} = \{a_1, \ldots, a_{|S_\ell|/2}\}$;
**9**      $\epsilon_{\ell+1} = 3\epsilon_\ell/4, \delta_{\ell+1} = \delta_\ell/2, \mu_{\ell+1} = (1 - \epsilon_\ell)\mu_\ell$, and $\ell = \ell + 1$;
**10 return** $S_\ell$;

---

**Algorithm 3:** Uniform Sampling (US)

**1 input:** $S, \epsilon, \delta, \mu_s, k$
**2** sample every arm $a \in S$ for $Q = (96/\epsilon^2)(1/\mu_s) \log(4|S|/\delta)$ times;
**3 for** *each arm* $a \in S$ **do**
**4**      its US-empirical value $\hat{\theta}^{US}(a)$ = the average of the $Q$ samples from $a$;
**5** $a_1, \ldots, a_{|S|}$ = the arms sorted in non-increasing order of their US-empirical values;
**6 return** $\{(a_1, \hat{\theta}^{US}(a_1)), \ldots, (a_k, \hat{\theta}^{US}(a_k))\}$

## 2   Top-$k$ Arm Selection

In this section, we describe a new algorithm for the $k$-AS problem. We present the detailed analysis in Appendix B.

Our $k$-AS algorithm consists of three components: ME-AS, *Median Elimination* (ME), and *Uniform Sampling* (US), as shown in Algorithms 1, 2, and 3, respectively.

Given parameters $B$, $\epsilon$, $\delta$, $k$ (as in Problem 1), ME-AS takes a "guess" $\mu$ (Line 2) on the value of $\theta_k(B)$, and then applies ME (Line 3) to prune $B$ down to a set $S$ of at most $4k$ arms. Then, at Line 4, US is invoked to process $S$. At Line 5, (as will be clear shortly) the value of $\hat{\theta}^{US}(a_k)$ is what ME-AS thinks should be the value of $\theta_k(B)$; thus, the algorithm performs a quality check to see whether $\hat{\theta}^{US}(a_k)$ is larger than but close to $\mu$. If the check fails, ME-AS halves its guess $\mu$ (Line 2), and repeats the above steps; otherwise, the output of US from Line 4 is returned as the final result.

ME runs in rounds. Round $\ell$ ($= 1, 2, ...$) is controlled by parameters $S_\ell, \epsilon_\ell, \delta_\ell$, and $\mu_\ell$ (their values for Round 1 are given at Line 1). In general, $S_\ell$ is the set of arms from which we still want to sample. For each arm $a \in S_\ell$, ME takes $Q_\ell$ (Line 4) samples from $a$, and calculates its *empirical value* $\hat{\theta}(a)$ (Lines 5 and 6). ME drops (at Lines 7 and 8) half of the arms in $S_\ell$ with the smallest empirical values, and then (at Line 9) sets the parameters of the next round. ME terminates by returning $S_\ell$ as soon as $|S_\ell|$ is at most $4k$ (Lines 3 and 10).

US simply takes $Q$ samples from each arm $a \in S$ (Line 2), and calculates its *US-empirical value* $\hat{\theta}^{US}(a)$ (Lines 3 and 4). Finally, US returns the $k$ arms in $S$ with the largest US-empirical values (Lines 5 and 6).

**Remark.** If we ignore Line 3 of Algorithm 1 and simply set $S = B$, then ME-AS degenerates into the algorithm in [14].

**Theorem 1** *ME-AS solves the $k$-AS problem with expected cost $O\left(\frac{n}{\epsilon^2}\frac{1}{\theta_k(B)}\log\frac{k}{\delta}\right)$.*

We extends the proof in [11] and establish the lower bound for $k$-AS as shown in Theorem 2.

**Theorem 2** *For any $\epsilon \in \left(0, \frac{1}{4}\right)$ and $\delta \in \left(0, \frac{1}{48}\right)$, given any algorithm, there is an instance of the $k$-AS problem on which the algorithm must entail $\Omega(\frac{n}{\epsilon^2}\frac{1}{\theta_k(B)}\log\frac{k}{\delta})$ cost in expectation.*

## 3    $k$-MOST CONNECTED VERTEX

This section is devoted to the $k$-MCV problem (Problem 3). We will focus on lower bounds because our $k$-AS algorithm in the previous section also settles $k$-MCV with the cost claimed in Table 1 by applying the reduction described in Section 1.1. We establish matching lower bounds below:

**Theorem 3** *For any $\epsilon \in \left(0, \frac{1}{12}\right)$ and $\delta \in \left(0, \frac{1}{48}\right)$, the following statements are true about any $k$-MCV algorithm:*

- *when $\deg_k(B) \geq \Omega\left(\frac{1}{\epsilon^2}\log\frac{n}{\delta}\right)$, there is an instance on which the algorithm must probe $\Omega(\frac{n}{\epsilon^2}\frac{m}{\deg_k(B)}\log\frac{k}{\delta})$ edges in expectation.*

- *when $\deg_k(B) < O(\frac{1}{\epsilon})$, there is an instance on which the algorithm must probe $\Omega(nm)$ edges in expectation.*

For large $\deg_k(B)$ in Theorem 3, we utilize an instance for $k$-AS to construct a random hidden bipartite graph and fed it to any algorithm solves $k$-MCV. By doing this, we reduce $k$-AS to $k$-MCV and thus, establish our first lower bound.

For small $\deg_k(B)$, we define the *single-vertex* problem where the goal is to distinguish two extremely distributions. We prove the lower bound of single-vertex problem and reduce it to $k$-MCV. Thus, we establish our second lower bound. The details are presented in Appendix D.

## 4    Top-$k_{\mathsf{avg}}$ Arm Selection

Our $k_{\mathsf{avg}}$-AS algorithm QE-AS is similar to ME-AS described in Section 2, except that the parameters are adjusted appropriately, as shown in Algorithm 4, 5, 6 respectively. We present the details in Appendix E.

**Theorem 4** *QE-AS solves the $k_{\mathsf{avg}}$-AS problem with expected cost $O\left(\frac{n}{\epsilon^2}\frac{1}{\theta_{\mathsf{avg}}(B)}\left(1+\frac{\log(1/\delta)}{k}\right)\right)$.*

We establish the lower bound for $k_{\mathsf{avg}}$-AS as shown in Theorem 5.

**Theorem 5** *For any $\epsilon \in \left(0, \frac{1}{12}\right)$ and $\delta \in \left(0, \frac{1}{48}\right)$, given any $(\epsilon, \delta)$-approximate algorithm, there is an instance of the $k_{\mathsf{avg}}$-AS problem on which the algorithm must entail $\Omega\left(\frac{n}{\epsilon^2}\frac{1}{\theta_{\mathsf{avg}}(B)}\left(1+\frac{\log(1/\delta)}{k}\right)\right)$ cost in expectation.*

We show that the lower bound of $k_{\mathsf{avg}}$-AS is the maximum of $\Omega\left(\frac{n}{\epsilon^2}\frac{1}{\theta_{\mathsf{avg}}(B)}\frac{\log(1/\delta)}{k}\right)$ and $\Omega\left(\frac{n}{\epsilon^2}\frac{1}{\theta_{\mathsf{avg}}(B)}\right)$. Our proof of the first lower bound is based on a novel reduction from 1-AS. We stress that our reduction can be used to simplify the proof of the lower bound in [16, Theorem 5.5].

## 5    $k_{\mathsf{avg}}$-MOST CONNECTED VERTEX

Our $k_{\mathsf{avg}}$-AS algorithm, combined with the reduction described in Section 1.1, already settles $k_{\mathsf{avg}}$-MCV with the sample complexity given in Table 1. We establish the following lower bound and prove it in Appendix F.

**Theorem 6** *For any $\epsilon \in \left(0, \frac{1}{12}\right)$ and $\delta \in \left(0, \frac{1}{48}\right)$, the following statements are true about any $k_{\mathsf{avg}}$-MCV algorithm:*

- *when $\deg_{\mathsf{avg}}(B) \geq \Omega\left(\frac{1}{\epsilon^2}\log\frac{n}{\delta}\right)$, there is an instance on which the algorithm must probe*

$$\Omega\left(\frac{n}{\epsilon^2}\frac{m}{\deg_{\mathsf{avg}}(B)}\left(1+\frac{\log(1/\delta)}{k}\right)\right)$$

  *edges in expectation.*

- *when $\deg_k(B) < O(\frac{1}{\epsilon})$, there is an instance on which the algorithm must probe $\Omega(nm)$ edges in expectation.*

**Algorithm 4:** QE-AS
---
1 **input:** $B, \epsilon, \delta, k$
2 **for** $\mu = 1/2, 1/4, \ldots$ **do**
3      $S = \text{QE}(B, \epsilon, \delta, \mu, k)$;
4      $\{(a_i \mid 1 \le i \le k), \hat{\theta}_{\text{avg}}^{US}\} = \text{US}(S, \epsilon, \delta, (1 - \epsilon/2)\mu, k)$;
5      **if** $\hat{\theta}_{\text{avg}}^{US} \ge 2\mu$ **then**
6          **return** $\{a_1, \ldots, a_k\}$;

---

**Algorithm 5:** Quartile Elimination (QE)
---
1 **input:** $B, \epsilon, \delta, \mu, k$
2 $S_1 = B, \epsilon_1 = \epsilon/32, \delta_1 = \delta/8, \mu_1 = \mu$, and $\ell = 1$;
3 **while** $|S_\ell| > 4k$ **do**
4      sample every arm $a \in S_\ell$ for $Q_\ell = (48/\epsilon_\ell^2)(1/\mu_\ell)\left(1 + \frac{\log(2/\delta_\ell)}{k}\right)$ times;
5      **for** *each arm* $a \in S_\ell$ **do**
6          its empirical value $\hat{\theta}(a)$ = the average of the $Q_\ell$ samples from $a$;
7      $a_1, \ldots, a_{|S_\ell|}$ = the arms sorted in non-increasing order of their empirical values;
8      $S_{\ell+1} = \{a_1, \ldots, a_{3|S_\ell|/4}\}$;
9      $\epsilon_{\ell+1} = 7\epsilon_\ell/8, \delta_{\ell+1} = \delta_\ell/2, \mu_{\ell+1} = (1 - \epsilon_\ell)\mu_\ell$, and $\ell = \ell + 1$;
10 **return** $S_\ell$;

---

**Algorithm 6:** Uniform Sampling (US)
---
1 **input:** $S, \epsilon, \delta, \mu_s, k$
2 sample every arm $a \in S$ for $Q = (120/\epsilon^2)(1/\mu_s)\left(1 + \frac{\log(4/\delta)}{k}\right)$ times;
3 **for** *each arm* $a \in S$ **do**
4      its US-empirical value $\hat{\theta}^{US}(a)$ = the average of the $Q$ samples from $a$;
5 $a_1, \ldots, a_{|S|}$ = the arms sorted in non-increasing order of their US-empirical values;
6 **return** $\{(a_1, \ldots, a_k), \hat{\theta}_{\text{avg}}^{US} = \frac{1}{k}\sum_{i=1}^{k} \hat{\theta}^{US}(a_i)\}$

## 6 Experiment Evaluation

Due to the space constraint, we show only the experiments that compare ME-AS and AMCV [14] for $k$-MCV problem. Additional experiments can be found in Appendix G. We use two synthetic data sets and one real world data set to evaluate the algorithms. Each dataset is represented as a bipartite graph with $n = m = 5000$. For the synthetic data, the degrees of the black vertices follow a power law distribution. For each black vertex $b \in B$, its degree equals $d$ with probability $c(d+1)^{-\tau}$ where $\tau$ is the parameter to be set and $c$ is the normalizing factor. Furthermore, for each black vertex with degree $d$, we connected it to $d$ randomly selected white vertices. Thus, we build two bipartite graphs by setting the proper parameters in order to control the average degrees of the black vertices to be 50 and 3000 respectively. For the real world data, we crawl 5000 active users from twitter with their corresponding relationships. We construct a bipartite graph $G = (B, W, E)$ where each of $B$ and $W$ represents all the users and $E$ represents the 2-hop relationships. We say two users $b \in B$ and $w \in W$ have a 2-hop relationship if they share at least one common friend.

As the theoretical analysis is rather pessimistic due to the extensive usage of the union bound, to make a fair comparison, we adopt the same strategy as in [14], i.e., to divide the sample cost in theory by a heuristic constant $\xi$. We use the same parameter $\xi = 2000$ for AMCV as in [14]. For ME-AS, we first take $\xi = 10^7$ for each round of the median elimination step and then we use the previous sample cost dividing 250 as the samples of the uniform sampling step. Notice that it does not conflict the theoretical sample complexity since the median elimination step dominates the sample complexity of the algorithm.

We fix the parameters $\delta = 0.1, k = 20$ and enumerate $\epsilon$ from 0.01 to 0.1. We then calculate the actual failure probability by counting the successful runs in 100 repeats. Recall that due to the heuristic nature, the algorithm may not achieve the theoretical guarantees prescribed by $(\epsilon, \delta)$.

Whenever this happens, we label the percentage of actual error $\epsilon_a$ it achieves according to the failure probability $\delta$. For example 2.9 means the algorithm actually achieves an error $\epsilon_a = 0.029$ with failure probability $\delta$. The experiment result is shown in Fig 1.

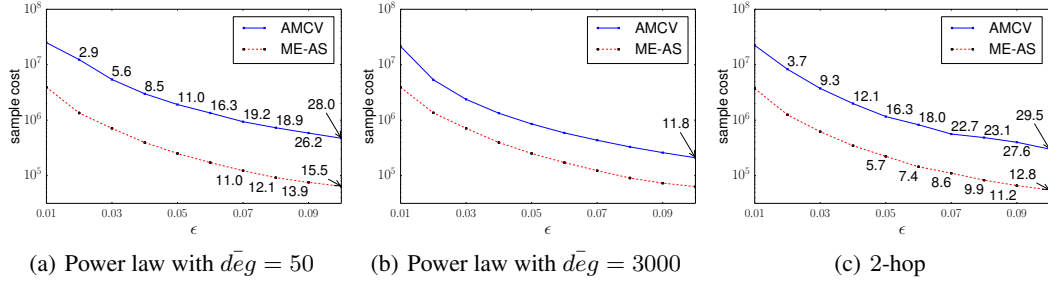

(a) Power law with $\bar{deg} = 50$    (b) Power law with $\bar{deg} = 3000$    (c) 2-hop

Figure 1: Performance comparison for $k$-MCV vs. $\epsilon$

As we can see, ME-AS outperforms AMCV in both sample complexity and the actual error in all data sets. We stress that in the worst case, it seems ME-AS only shows a difference when $n \gg k$. However for the most of the real world data, the degrees of the vertices usually follow a power law distribution or a Gaussian distribution. For such cases, our algorithm only needs to take a few samples in each round of the elimination step and drops half of vertices with high confidence. Therefore, the experimental result shows that the sample cost of ME-AS is much less than AMCV.

## 7 Related Work

Multi-armed bandit problems are classical decision problems with exploration-exploitation trade-offs, and have been extensively studied for several decades (dating back to 1930s). In this line of research, $k$-AS and $k_{\mathsf{avg}}$-AS fit into the *pure exploration* category, which has attracted significant attentions in recent years due to its abundant applications such as online advertisement placement [6], channel allocation for mobile communications [2], crowdsourcing [16], etc. We mention some closely related work below, and refer the interested readers to a recent survey [4].

Even-Dar et al. [8] proposed an optimal algorithm for selecting a single arm which approximates the best arm with an additive error at most $\epsilon$ (a matching lower bound was established by Mannor et al. [12]). Kalyanakrishnan et al. [10, 11] considered the EXPLORE-$k$ problem which we mentioned in Section 1.2. They provided an algorithm with the sample complexity $O(\frac{n}{\epsilon^2} \log \frac{k}{\delta})$. Similarly, Zhou et al. [16] studied the OPTMAI problem which, again as mentioned in Section 1.2, is the absolute-error version of $k_{\mathsf{avg}}$-AS.

Audibert et al. [2] and Bubeck et al. [4] investigated the *fixed budget* setting where, given a fixed number of samples, we want to minimize the so-called *misidentification probability* (informally, the probability that the solution is not optimal). Buckeck et al. [5] also showed the links between the simple regret (the gap between the arm we obtain and the best arm) and the cumulative regret (the gap between the reward we obtained and the expected reward of the best arm). Gabillon et al. [9] provide a unified approach UGapE for EXPLORE-$k$ in both the fixed budget and the fixed confidence settings. They derived the algorithms based on "lower and upper confidence bound" (LUCB) where the time complexity depends on the gap between $\theta_k(B)$ and the other arms . Note that each time LUCB samples the two arms that are most difficult to distinguish. Since our problem ensures an individually guarantee, it is unclear whether only sampling the most difficult-to-distinguish arms would be enough. We leave it as an intriguing direction for future work. Chen et al. [6] studied how to select the best arms under various combinatorial constraints.

**Acknowledgements.** Jian Li, Wei Cao, Zhize Li were supported in part by the National Basic Research Program of China grants 2015CB358700, 2011CBA00300, 2011CBA00301, and the National NSFC grants 61202009, 61033001, 61361136003. Yufei Tao was supported in part by projects GRF 4168/13 and GRF 142072/14 from HKRGC.

## Footnotes

[1]The algorithm was designed for $k$-MCV, but it can be adapted to $k$-AS as well.

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
