[Supplementary Material]

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

## Organization

The appendix is organized as follows. We first review several useful forms of Chernoff bounds in Appendix A. In Appendix B, we analyze our algorithm for $k$-AS and the corresponding lower bound (Theorem 1 and 2). We provide the algorithm and the analysis for $k_{\mathsf{add}}$-AS in Section C. In Appendix D, we provide the deferred proof of our lower bound for $k$-MCV. Next, we provide the deferred analysis of algorithm and the proof of the lower bound for $k_{\mathsf{avg}}$-AS (Theorem 4 and 5) in Appendix E. In Appendix F, we prove our lower bound for $k_{\mathsf{avg}}$-MCV (Theorem 6). Finally, we present the additional experiment evaluation in Appendix G.

## A  Chernoff Bounds

**Proposition 1 ([7])** *Let $X_1, \ldots, X_n$ be independent random variables with $\Pr[X_i = 1] = p_i$, and $\Pr[X_i = 0] = 1 - p_i$. Define $X = \sum_{i=1}^{n} X_i$. Clearly, $\mathbb{E}[X] = \sum_{i=1}^{n} p_i$. Then, for $0 < \alpha < 1$:*

$$\Pr[X \geq (1+\alpha)\mathbb{E}(X)] \leq \exp\left(\frac{-\alpha^2 \mathbb{E}(X)}{3}\right). \tag{1}$$

$$\Pr[X \leq (1-\alpha)\mathbb{E}(X)] \leq \exp\left(\frac{-\alpha^2 \mathbb{E}(X)}{2}\right). \tag{2}$$

*For $\alpha \geq 1$:*

$$\Pr[X \geq (1+\alpha)\mathbb{E}(X)] \leq \exp\left(\frac{-(1+\alpha)\mathbb{E}(X)}{6}\right). \tag{3}$$

*For $\alpha > 0$:*

$$\Pr[X \geq (1+\alpha)\mathbb{E}(X)] \leq \left(\frac{e}{1+\alpha}\right)^{(1+\alpha)\mathbb{E}(X)}. \tag{4}$$

$$\Pr[X \leq \mathbb{E}(X) - \alpha] \leq \exp\left(-\frac{\alpha^2}{2\mathbb{E}(X)}\right). \tag{5}$$

$$\Pr[|X - \mathbb{E}(X)| \geq \alpha] \leq 2\exp(-2\alpha^2 n). \tag{6}$$

*Moreover, suppose $\mathbb{E}[X_i] < a$ for some real $0 \leq a \leq 1$. For every $t > 0$, we have that*

$$\Pr[X/n > a + t] < ((\frac{a}{a+t})^{a+t}(\frac{1-a}{1-a-t})^{1-a-t})^n \tag{7}$$

## B  Analysis of Top-$k$ Arm Selection

In this section we analyze our $k$-AS algorithm. We first analyze the Median Elimination in Appendix B.1. In Appendix B.2, we analyze the correctness and the sample complexity which prove Theorem 1. Then, we prove Theorem 2 in Appendix B.3.

### B.1  Analysis of Median Elimination

This subsection serves as a proof for the following lemma, which says that if the guess $\mu$ (passed in by ME-AS) is low enough, with a large probability, ME returns a good arm set $S$ that preserves approximately the $k$ best arms in $B$:

**Lemma 1** *In Median Elimination, if $\mu \leq \theta_k(B)$, with probability at least $1 - \frac{\delta}{4}$, ME returns an arm set $S$ satisfying $\theta_i(S) \geq \left(1 - \frac{\epsilon}{2}\right)\theta_i(B)$ for all $i \leq k$.*

Let us first focus on the $\ell$-th round for a specific $\ell$. Recall that, in this round, ME calculates an empirical value for each arm in $S_\ell$. Henceforth, for any subset $X \subseteq S_\ell$, define $\hat{\theta}_i(X)$ to be the empirical value of $a_i(X)$.

Let $\mathcal{E}_\ell$ be the event where the following holds for *all* $i \leq k$:

$$\hat{\theta}_i(S_\ell) \in \left[(1 - \frac{\epsilon_\ell}{2})\theta_i(S_\ell), (1 + \frac{\epsilon_\ell}{2})\theta_i(S_\ell)\right].$$

Moreover, we say that an arm $a \in S_\ell$ is a *bad arm* if there exists an $i \leq k$ such that $\theta(a) < (1 - \epsilon_\ell)\theta_i(S_\ell)$ whereas $\hat{\theta}(a) \geq (1 - \frac{\epsilon_\ell}{2})\theta_i(S_\ell)$.

**Lemma 2** *In Round $\ell$, if $\mathcal{E}_\ell$ occurs and the number of bad arms is no more than $\frac{|S_\ell|}{2} - k$, then $\theta_i(S_{\ell+1}) \geq (1 - \epsilon_\ell)\theta_i(S_\ell)$ holds for all $i \leq k$.*

*Proof*: Under $\mathcal{E}_\ell$, no arm $a_i(S_\ell)$ of any $i \leq k$ can be bad. To see this, suppose that there exists $j < i$ satisfying $\theta_i(S_\ell) < (1 - \epsilon_\ell)\theta_j(S_\ell)$ but $\hat{\theta}_i(S_\ell) \geq (1 - \frac{\epsilon_\ell}{2})\theta_j(S_\ell)$. From $\mathcal{E}_\ell$, we have $\hat{\theta}_i(S_\ell) \leq (1 + \frac{\epsilon_\ell}{2})\theta_i(S_\ell) < (1 + \frac{\epsilon_\ell}{2})(1 - \epsilon_\ell)\hat{\theta}_j(S_\ell) < (1 - \frac{\epsilon_\ell}{2})\theta_j(S_\ell)$, which is a contradiction.

When the number of bad arms is at most $\frac{|S_\ell|}{2} - k$, at least $k$ arms in $S_{\ell+1}$ are not bad. Denote the set of these arms as $T$. The rest of the proof will establish the claim:

$$\theta_i(T) \geq (1 - \epsilon_\ell)\theta_i(S_\ell), \forall i \leq k$$

which is sufficient for the lemma to hold.

Suppose that the claim does not hold. Consider the smallest $i \leq k$ such that $\theta_i(T) < (1 - \epsilon_\ell)\,\theta_i(S_\ell)$. Since $a_i(T)$ is not bad, it holds that $\hat{\theta}_i(T) < (1 - \frac{\epsilon_\ell}{2})\theta_i(S_\ell)$. Event $\mathcal{E}_\ell$ suggests that $\hat{\theta}_i(S_\ell) \geq (1 - \frac{\epsilon_\ell}{2})\theta_i(S_\ell) > \hat{\theta}_i(T)$ which implies that $a_i(S_\ell)$ has not been eliminated in this round. This, together with the fact (shown earlier) that $a_i(S_\ell)$ is not bad, indicates that $a_i(S_\ell)$ must match an arm $a_{i'}(T)$ with $i' \leq i - 1$.

For any $j < i$, since $\theta_j(S_\ell) \geq \theta_i(S_\ell)$, we know that $\theta_i(T) < (1 - \epsilon_\ell)\theta_j(S_\ell)$. A similar argument shows that $a_j(S_\ell)$ matches an arm $a_{j'}(T)$ with $j' \leq i - 1$ and yet $j' \neq i'$. This means that the first $i$ elements of $S_\ell$ form a one-to-one mapping to the first $i - 1$ elements of $T$, which is a contradiction. □

**Lemma 3** *Under the condition $\mu \leq \theta_k(B)$, if for all $j < \ell$, Median Elimination ensures*

$$\theta_i(S_{j+1}) \geq (1 - \epsilon_j)\,\theta_i(S_j), \forall i \leq k \tag{8}$$

*then, with probability at least $1 - \delta_\ell$, $\theta_i(S_{\ell+1}) \geq (1 - \epsilon_\ell)\theta_i(S_\ell)$ holds for all $i \leq k$.*

*Proof*: By Lemma 2, it suffices to show that with probability no less than $1 - \delta_\ell$, $\mathcal{E}_\ell$ occurs and the number of bad arms does not exceed $\frac{|S_\ell|}{2} - k$.

Recall that in ME $\theta_k(S_1) = \theta_k(B) \geq \mu = \mu_1$. Therefore, from (8), we know $\theta_k(S_l) \geq \prod_{j=1}^{l-1}(1 - \epsilon_j)\,\theta_k(S_1) \geq \mu_l$. Thus, by Chernoff bounds (1) and (2), we have that for any $i \leq k$:

$$\Pr\left[\hat{\theta}_i(S_\ell) \leq \left(1 - \frac{\epsilon_\ell}{2}\right)\theta_i(S_\ell)\right]$$
$$\leq \quad \exp\left(-\frac{(\epsilon_\ell/2)^2}{2}Q_\ell\theta_i(S_\ell)\right) \leq \frac{\delta_\ell}{6k},$$

and also

$$\Pr\left[\hat{\theta}_i(S_\ell) \geq \left(1 + \frac{\epsilon_\ell}{2}\right)\theta_i(S_\ell)\right] \leq \frac{\delta_\ell}{6k}.$$

By the union bound, the failure probability of $\mathcal{E}_\ell$ is at most $\delta_\ell/3$.

Next, consider the probability of an arm $a$ being bad. For any $i \leq k$ which satisfies $\theta(a) < (1 - \epsilon_\ell)\,\theta_i(S_\ell)$, define $\alpha = \frac{\theta_i(S_\ell)}{\theta(a)}\left(1 - \frac{\epsilon_\ell}{2}\right) - 1$. We distinguish two cases:

- *Case $0 < \alpha < 1$.* From Chernoff bound (1),

$$\Pr\left[\hat{\theta}(a) \geq \left(1 - \frac{\epsilon_\ell}{2}\right)\theta_i(S_\ell)\right]$$
$$= \quad \Pr\left[\hat{\theta}(a) \geq (1 + \alpha)\,\theta(a)\right]$$
$$\leq \quad \exp\left(-\frac{(1 - \epsilon_\ell/2)}{3}Q_\ell\theta_i(S_\ell) \cdot \frac{\alpha^2}{1 + \alpha}\right).$$

Recall that $\alpha = \frac{\theta_i(S_\ell)}{\theta(a)}\left(1 - \frac{\epsilon_\ell}{2}\right) - 1 \geq \frac{1}{1-\epsilon_\ell}\left(1 - \frac{\epsilon_\ell}{2}\right) - 1 = \frac{\epsilon_\ell}{2-2\epsilon_\ell}$ and $\frac{\alpha^2}{1+\alpha}$ increases monotonically when $\alpha > 0$. Therefore, we have

$$\Pr\left[\hat{\theta}(a) \geq (1+\alpha)\,\theta(a)\right]$$

$$\leq \quad \exp\left(-\frac{(1-\epsilon_\ell/2)}{3}Q_\ell\theta_i(S_\ell) \cdot \frac{\epsilon_\ell^2}{(2-\epsilon_\ell)(2-2\epsilon_\ell)}\right)$$

$$\leq \quad \exp\left(-\frac{(\epsilon_\ell/2)^2}{3}Q_\ell\theta_i(S_\ell)\right) \leq \frac{\delta_\ell}{6k}.$$

- *Case $\alpha \geq 1$.* From Chernoff bound (3), we know

$$\Pr\left[\hat{\theta}(a) \geq (1+\alpha)\,\theta(a)\right]$$

$$\leq \quad \exp\left(-\frac{\left(1 - \frac{\epsilon_\ell}{2}\right)}{6}Q_\ell\theta_i(S_\ell)\right)$$

$$\leq \quad \exp\left(-\frac{(\epsilon_\ell/2)^2}{2}Q_\ell\theta_i(S_\ell)\right) \leq \frac{\delta_\ell}{6k},$$

where the second last inequality holds due to the fact that $\frac{\left(1-\frac{\epsilon}{2}\right)}{6} \geq \frac{(\epsilon/2)^2}{2}$ for any $\epsilon \in \left(0, \frac{1}{4}\right)$.

By the union bound, the probability of the arm $a$ being bad is at most $\frac{\delta_\ell}{6}$.

Let $Z_i$ be a Boolean random variable that equals 1 if $a_i(S_\ell)$ is a bad arm, and 0 otherwise. By Markov's inequality, we have that

$$\Pr\left[\sum_{i=1}^{|S_\ell|} Z_i > \frac{|S_\ell|}{2} - k\right] \quad \leq \quad \frac{\mathbb{E}\left[\sum_{i=1}^{|S_\ell|} Z_i\right]}{|S_\ell|/2 - k}$$

$$\leq \quad \frac{|S_\ell|\delta_\ell/6}{|S_\ell|/2 - k} \leq \frac{2\delta_\ell}{3},$$

where the last inequality holds because $|S_\ell| > 4k$ for any $\ell$.

By the union bound, we conclude that with probability at least $1 - \delta_\ell$, $\theta_i(S_{\ell+1}) \geq (1-\epsilon_\ell)\theta_i(S_\ell)$ holds for all $i \leq k$. $\qquad\square$

We are now ready to prove Lemma 1. Suppose that ME terminates after $L$ rounds. Lemma 3 suggests that, with probability at least

$$1 - \sum_{i=1}^{L}\delta_i = 1 - \sum_{i=1}^{L}\frac{\delta_1}{2^{i-1}} \geq 1 - 2\delta_1 = 1 - \delta/4,$$

$\theta_i(S_{L+1}) \geq \theta_i(B)\prod_{j=1}^{L}(1-\epsilon_j)$ holds for all $i \leq k$. Utilizing the general fact that $-2\epsilon \leq \log(1-\epsilon) \leq -\epsilon$ is true for any $\epsilon \in \left(0, \frac{1}{4}\right)$, we have:

$$\prod_{j=1}^{L}(1-\epsilon_j) \quad = \quad \exp\left(\sum_{j=1}^{L}\log(1-\epsilon_j)\right)$$

$$\geq \quad \exp\left(-2\epsilon_1\sum_{j=1}^{\infty}\left(\frac{3}{4}\right)^{j-1}\right)$$

$$= \quad \exp(-8\epsilon_1) = \exp(-\epsilon/2)$$

$$\geq \quad 1 - \epsilon/2.$$

This completes the proof of Lemma 1.

### B.2 Correctness and Sample Complexity

**Analysis of Uniform Sampling.** As mentioned in Section 2, US was already applied in the algorithm developed in by Sheng et al. [14]. We can re-use much of their analysis in our context. First, they proved:

**Lemma 4 ([14])** *In Uniform Sampling, if $\mu_s \leq \theta_k(S)$, then with probability at least $1 - \frac{\delta}{4}$, US returns an arm set $V$ which satisfies $\theta_i(V) \geq \left(1 - \frac{\epsilon}{2}\right)\theta_i(S)$ for any $i \leq k$.*

*Proof*: By the proof of Lemma 8 in [14] and adjusting constants appropriately. $\square$

Next, we show that with a large probability ME-AS terminates at $\mu \in [\frac{1}{8}\theta_k(B), \theta_k(B)]$. If this is not what happens, one of the following two events must have happened:

    (1) *Premature*: ME-AS terminates when $\mu > \theta_k(B)$.

    (2) *Overdue*: ME-AS does not terminate after invoking US with $\mu < \frac{1}{4}\theta_k(B)$.

We say that ME *succeeds* if the arm set $S$ it returns satisfies $\theta_i(S) \geq (1 - \frac{\epsilon}{2})\theta_i(B)$ for all $i \geq k$.

**Lemma 5** *Both of the following statements are true:*

- *(Due to [14]) The premature event happens with probability at most $\delta/4$.*

- *If ME succeeds, the overdue event happens with probability at most $\delta/4$.*

*Proof*: The first statement has been proved in [14] (see Lemma 9 therein and adjusting constants appropriately). For the overdue event, when $\mu < \theta_k(B)/4$, given that ME has succeeded, we have $\mu_s = (1 - \frac{\epsilon}{2})\mu < \theta_k(S)$. Define $\hat{\theta}_k^{US}(S)$ to be the US-empirical value of $a_k(S)$. We have:

$$
\begin{aligned}
\Pr\left[\hat{\theta}_k^{US}(S) \leq 2\mu\right] &\leq \Pr\left[\hat{\theta}_k^{US}(S) \leq \theta_k(B)/2\right] \\
&\leq \Pr\left[\hat{\theta}_k^{US}(S) \leq \frac{\theta_k(S)}{2(1 - \epsilon/2)}\right] \\
&\leq \Pr\left[\hat{\theta}_k^{US}(S) \leq (1 - \epsilon/2)\theta_k(S)\right] \\
\text{(by (2))} \quad &\leq \delta/4,
\end{aligned}
$$

where the third inequality used the fact that $\frac{1}{2(1-\epsilon/2)} \leq (1 - \frac{\epsilon}{2})$ holds for any $\epsilon \in (0, \frac{1}{4})$. $\square$

**Correctness of ME-AS.** Combining Lemmas 1, 4, and 5, we now prove:

**Lemma 6** *With probability at least $1 - \delta$, ME-AS returns an arm set $V$ which satisfies $\theta_i(V) \geq (1 - \epsilon)\theta_i(B)$ for all $i \leq k$.*

*Proof*: If the premature event does not occur, we have that $\mu \leq \theta_k(B)$. Provided that ME has succeeded, Lemma 5 has showed that, with probability at least $1 - \frac{\delta}{4}$, ME-AS terminates at $\mu \in [\frac{1}{8}\theta_k(B), \theta_k(B)]$. ME having succeeded ensures $\theta_i(S) \geq (1 - \frac{\epsilon}{2})\theta_i(B)$ for all $i \leq k$. Hence, $\mu_s \leq \theta_k(S)$. Then, by Lemma 4, we know that with probability at least $1 - \frac{\delta}{4}$, the following holds for all $i \leq k$:

$$
\begin{aligned}
\theta_i(V) &\geq (1 - \epsilon/2)\theta_i(S) \\
&\geq (1 - \epsilon/2)^2\theta_i(B) \geq (1 - \epsilon)\theta_i(B).
\end{aligned}
$$

The premature event occurs with probability at most $\frac{\delta}{4}$. By Lemma 1, ME fails with probability only at most $\frac{\delta}{4}$ when $\mu \leq \theta_k(B)$. Lemma 6 thus follows from the union bound. $\square$

**Sample Complexity of ME-AS.** We first analyze the cost of ME and US under a specific $\mu$. This is trivial for US, for which the answer is obviously $O(\frac{k}{\epsilon^2}\frac{1}{\mu}\log\frac{k}{\delta})$ (recall that $|S| \leq 4k$).

Regarding the $\ell$-th round of ME, we know $|S_l| = \frac{n}{2^{l-1}}$, $\epsilon_\ell = (3/4)^{\ell-1}\epsilon/16$, $\delta_\ell = (1/2)^{\ell-1}\delta/8$, and $\mu_\ell = \prod_{j=1}^{l-1}(1-\epsilon_j)u_1 \geq (1-\epsilon/2)\mu \geq 7/(8\mu)$. Hence, the cost of ME is bounded by:

$$\sum_{\ell=1}^{\infty} \frac{12}{\epsilon_\ell^2} \frac{1}{\mu_\ell} \log\left(\frac{6k}{\delta_\ell}\right) |S_\ell|$$

$$= O\left(\sum_{\ell=1}^{\infty} \frac{12}{\left(\left(\frac{3}{4}\right)^{\ell-1}\frac{\epsilon}{16}\right)^2}\frac{1}{\mu}\left(\log\frac{k}{\delta}+\ell+1\right)\frac{n}{2^{\ell-1}}\right)$$

$$= O\left(\sum_{\ell=1}^{\infty}\left(\frac{8}{9}\right)^{\ell-1}\frac{n}{\epsilon^2}\frac{1}{\mu}\left(\log\frac{k}{\delta}+\ell\right)\right)$$

$$= O\left(\frac{n}{\epsilon^2}\frac{1}{\mu}\log\frac{k}{\delta}\right).$$

We have proved that one for-iteration of ME-AS (i.e., Lines 3-6 of Algorithm 1) under a specific $\mu$ has cost $O(\frac{k}{\epsilon^2}\frac{1}{\mu}\log\frac{k}{\delta})$. The cost doubles each time when we halve the parameter $\mu$, until $\mu$ drops below $\theta_k(B)/4$. Hence, the total complexity so far is $O(\frac{n}{\epsilon^2}\frac{1}{\theta_k(B)}\log\frac{k}{\delta})$.

It remains to show that if ME-AS terminates at $\mu < \theta_k(B)/8$, the extra cost is still bounded by $O(\frac{n}{\epsilon^2}\frac{1}{\theta_k(B)}\log\frac{k}{\delta})$ in expectation. From Lemmas 1 and 5, we know that, at every $\mu < \theta_k(B)/8$, ME-AS fails to terminate with probability at most $\delta/2$. Thus, its cost for *all* $\mu < \theta_k(B)/8$ is bounded by

$$O\left(\frac{n}{\epsilon^2}\frac{1}{\theta_k(B)}\log\frac{k}{\delta}\right)\sum_{i=1}^{\infty}\left(2^{i-1}\right)\left(\frac{\delta}{2}\right)^{i-1}$$

$$= O\left(\frac{n}{\epsilon^2}\frac{1}{\theta_k(B)}\log\frac{k}{\delta}\right).$$

We thus have proved Theorem 1.

**Theorem 1** *ME-AS solves the $k$-AS problem with expected cost $O\left(\frac{n}{\epsilon^2}\frac{1}{\theta_k(B)}\log\frac{k}{\delta}\right)$.*

## B.3 Lower Bound

This subsection serves as a proof for:

**Theorem 2** *For any $\epsilon \in \left(0, \frac{1}{4}\right)$ and $\delta \in \left(0, \frac{1}{48}\right)$, given any algorithm, there is an instance of the $k$-AS problem on which the algorithm must entail $\Omega(\frac{n}{\epsilon^2}\frac{1}{\theta_k(B)}\log\frac{k}{\delta})$ cost in expectation.*

**Hard Instances.** We use $a_1, \ldots, a_n$ to represent the $n$ arms in $B$. Let $\mathcal{I}_x$ denote the collection of all the $x$-sized subset of $U = \{a_2, \ldots, a_n\}$. For each $I \in \mathcal{I}_k \cup \mathcal{I}_{k-1}$, we create an instance with:

$$\theta(a_1) = (1+4\epsilon)\theta \tag{9}$$
$$\theta(a_i) = (1+8\epsilon)\theta, \forall a_i \in I \tag{10}$$
$$\theta(a_i) = \theta, \forall a_i \in U \setminus I \tag{11}$$

where $\theta$ is an arbitrary real value in $(0, \frac{1}{12})$.

We use $\Theta_I$ to denote the instance associated with $I$. Notice for $\epsilon \in \left(0, \frac{1}{4}\right)$, $(1-\epsilon)(1+4\epsilon)\theta > \theta$ and $(1-\epsilon)(1+8\epsilon)\theta > (1+4\epsilon)\theta$. Thus, by the problem definition of $k$-AS, any algorithm, with probability at least $1-\delta$, must return $\{a_1\} \cup I$ for $I \in \mathcal{I}_{k-1}$ and return $I$ for $I \in \mathcal{I}_k$.

**Deterministic Algorithms.** We will first consider an algorithm $\mathcal{A}$ deterministic in the following sense. At the beginning, the first arm that $\mathcal{A}$ samples from is fixed. From then on, iteratively, the next arm to sample from is always a function of the results of the previous samples (i.e., which arms have been sampled, in what order, and what are the results). In this way, $\mathcal{A}$ can be uniquely

described by a *history* $W$, which is the sequence of arms chosen at each step, and the result of each sample. Finally, the output of $\mathcal{A}$ is a function of $W$.

It is important to note that $W$ is a random variable, due to the randomness from the arms, namely, the samples taken from them are random. Even on a specific problem instance, various histories $W$ can occur with different probabilities. Therefore, the cost of $\mathcal{A}$ (i.e., the length of $W$) is still a random variable; and also $\mathcal{A}$ may choose to fail on some $W$, provided that the overall failure probability on the instance is at most $\delta$.

The crux of our proof is to show that, for any deterministic algorithm $\mathcal{A}$, there exists $I \in \mathcal{I}_{k-1}$ such that $\mathcal{A}$ must take at least $\Omega\left(\frac{n}{\epsilon^2}\frac{1}{\theta}\ln\frac{k}{\delta}\right)$ samples in expectation to solve the $k$-AS problem on instance $\Theta_I$. Assume, on the contrary, that for any $I \in \mathcal{I}_{k-1}$, $\mathcal{A}$ takes at most $\frac{n}{36864\epsilon^2}\frac{1}{\theta}\ln\frac{k}{\delta}$ samples in expectation. We will prove that $\mathcal{A}$ must fail with a probability greater than $\delta$ on some instance $\Theta_I$ with $I \in \mathcal{I}_k$.

**A Deterministic Lower Bound.** Set $T = \frac{1}{2304\epsilon^2}\frac{1}{\theta}\ln\frac{k}{\delta}$. Define random variable $T_j$ be the number of samples that $\mathcal{A}$ takes from arm $a_j$. It is easy to see that $T_j > 0$ (i.e., $\mathcal{A}$ must sample at least once from $a_j$).

Given a specific $I \in \mathcal{I}_{k-1}$, denote by $\mathbb{E}_I[T_j]$ the expectation of $T_j$ when $\mathcal{A}$ runs on $\Theta_I$. At most $n/4$ arms $a_j \in U \setminus I$ satisfy

$$\mathbb{E}_I[T_j] > \frac{1}{9216\epsilon^2}\frac{1}{\theta}\ln\frac{k}{\delta}.$$

Since $k \leq n/2$, at least $n - k - \frac{n}{4} \geq \frac{n-k}{2}$ arms $a_j \in U \setminus I$ satisfy $\mathbb{E}_I[T_j] \leq \frac{1}{9216\epsilon^2}\frac{1}{\theta}\ln\frac{k}{\delta}$. We refer to these arms as *ordinary arms*. For any such an arm $a_j$, by Markov's inequality, we know:

$$\Pr_I[T_j > T] < \frac{\mathbb{E}_I[T_j]}{T} \leq \frac{1}{4}, \tag{12}$$

where the subscript $I$ of $\Pr_I$ (just like the subscript in $\mathbb{E}_I$) indicates "conditioned on the specific instance $I$".

Now let us fix an ordinary arm $a_j$. Recall that each of the $T_j$ samples that $\mathcal{A}$ draws from $a_j$ is either 1 or 0. Let $X$ be the sum of all those $T_j$ samples. Set $\Delta = \sqrt{\theta T \ln(k/\delta)}$ and define $\mathcal{E}_A$ be the event where both $T_j \leq T$ and $X > \theta T_j - \Delta$ are true.

**Lemma 7** $\mathcal{E}_A$ *happens with probability at least* $1/2$.

*Proof*: By substituting $\theta(a_j) = \theta$, we can upper bound the probability of $\mathcal{E}_A$ *not* happening by:

$$\Pr_I[T_j > T] + \Pr_I[T_j \leq T, X - \theta T_j \leq -\Delta]$$

$$\leq \quad \frac{1}{4} + \Pr_I\left[T_j \leq T, X - \theta T_j \leq -\sqrt{\theta T_j \ln(k/\delta)}\right]$$

$$= \quad \frac{1}{4} + \sum_{t=1}^{T}\Pr[T_j = t]\cdot\Pr\left[X - \theta t \leq -\sqrt{\theta t \ln(k/\delta)}\right]$$

$$\leq \quad \frac{1}{4} + \sum_{t=1}^{T}\Pr[T_j = t]\cdot\exp\left(-\frac{1}{2}\ln\frac{k}{\delta}\right)$$

$$= \quad \frac{1}{4} + \exp\left(-\frac{1}{2}\ln\frac{k}{\delta}\right) = \frac{1}{4} + \left(\frac{\delta}{k}\right)^{1/2} \leq \frac{1}{2}$$

where the first inequality used (12), and the second inequality is due to Chernoff bound (5). □

Let $V$ be the arm set returned by $\mathcal{A}$ and $\mathcal{E}_B$ be the event that $V = I \cup \{a_1\}$. Define event $\mathcal{E}_S = \mathcal{E}_A \cap \mathcal{E}_B$. As $\mathcal{A}$ (when executed on $\Theta_I$) returns $I \cup \{a_1\}$ with probability at least $1 - \delta$, we know that

$$\Pr_I[\mathcal{E}_S] \geq 1 - \delta - 1/2 \geq 23/48. \tag{13}$$

Let us focus on one specific history $W$. Consider the instance $I \cup \{a_j\} \in \mathcal{I}_k$, which can be conveniently understood as raising the mean of arm $a_j$ from $\theta$ to $(1 + 8\epsilon)\theta$ in the instance $I$. Denote by $\Pr_{I \cup \{a_j\}}[W]$ the probability of $W$ when $\mathcal{A}$ runs on $\Theta_{I \cup \{a_j\}}$, and similarly, by $\Pr_I[W]$ the probability of $W$ on $\Theta_I$.

**Lemma 8** *For any $\epsilon \in (0, \frac{1}{4}), \delta \in (0, \frac{1}{48})$, when conditioned on $\mathcal{E}_S$, it holds that*

$$\frac{\Pr_{I \cup \{a_j\}}[W]}{\Pr_I[W]} \geq \frac{5\delta}{k}.$$

*Proof*: As the mean of $a_j$ is larger under $\Theta_{I \cup \{a_j\}}$, the ratio of the two probabilities is minimized when the fewest samples on $a_j$ turn out to be 1 in $W$. Recall that the event $\mathcal{E}_S$ ensures $X > \theta T_j - \Delta$. With this, we can derive

$$\frac{\Pr_{I \cup \{a_j\}}[W]}{\Pr_I[W]}$$

$$\geq \frac{[(1 + 8\epsilon)\theta]^{\theta T_j - \Delta}}{\theta^{\theta T_j - \Delta}} \frac{[1 - (1 + 8\epsilon)\theta]^{(1-\theta)T_j + \Delta}}{(1 - \theta)^{(1-\theta)T_j + \Delta}}$$

$$= \left[ (1 + 8\epsilon) \left( \frac{1 - (1 + 8\epsilon)\theta}{1 - \theta} \right)^{\frac{1-\theta}{\theta}} \right]^{\theta T_j - \Delta}$$

$$\left( \frac{1 - (1 + 8\epsilon)\theta}{1 - \theta} \right)^{\frac{\Delta}{\theta}}$$

$$\geq \left[ (1 + 8\epsilon) \left( \frac{1 - (1 + 8\epsilon)\theta}{1 - \theta} \right)^{\frac{1-\theta}{\theta}} \right]^{\theta T_j - \Delta}$$

$$(1 - 12\epsilon\theta)^{\frac{\Delta}{\theta}} , \tag{14}$$

where the last inequality used the fact that $1 - (1 + 8\epsilon)\theta \geq (1 - \theta)(1 - 12\epsilon\theta)$ for any $\theta \in \left( 0, \frac{1}{12} \right)$.

For any $x \in \left( 0, \frac{1}{4} \right)$, it holds that $e^{-1} \geq (1 - x)^{\frac{1}{x}} \geq e^{-2}$. Using this fact and plugging in $\Delta = \sqrt{\theta T \ln(k/\delta)}$ and $T = \frac{1}{2304\epsilon^2} \frac{1}{\theta} \ln \frac{k}{\delta}$, we can bound the second term of (14) as:

$$(1 - 12\epsilon\theta)^{\frac{\Delta}{\theta}} = (1 - 12\epsilon\theta)^{\frac{1}{48\epsilon\theta} \ln \frac{k}{\delta}}$$

$$\geq e^{-\frac{1}{2} \ln(k/\delta)} = \sqrt{\delta/k}. \tag{15}$$

To bound the first term of (14), let us define $g(\theta) = \left( \frac{1 - (1 + 8\epsilon)\theta}{1 - \theta} \right)^{\frac{1-\theta}{\theta}}$. Taking the partial derivative of $\ln g(\theta)$ with respect to $\theta$ gives:

$$\frac{\partial \ln g(\theta)}{\partial \theta} = -\frac{1}{\theta^2} \ln \left( 1 - 8\epsilon \frac{\theta}{1 - \theta} \right)$$

$$- \left( \frac{1}{\theta} - 1 \right) \frac{8\epsilon}{1 - 8\epsilon \frac{\theta}{1-\theta}} \frac{1}{(1 - \theta)^2}. \tag{16}$$

It is not hard to verify that for any $\theta \in \left( 0, \frac{1}{12} \right), \epsilon \in \left( 0, \frac{1}{4} \right)$, the RHS of (16) is strictly less than 0, which implies

$$g(\theta) \geq g \left( \frac{1}{12} \right) = e^{\ln\left( 1 - \frac{8}{11}\epsilon \right)^{11}} \geq e^{\ln(1 - 8\epsilon)} = 1 - 8\epsilon.$$

Moreover, observe that

$$(1 + 8\epsilon) \left( \frac{1 - (1 + 8\epsilon)\theta}{1 - \theta} \right)^{\frac{1-\theta}{\theta}}$$

$$= (1 + 8\epsilon) \left( 1 - 8\epsilon \frac{\theta}{1 - \theta} \right)^{\frac{1-\theta}{\theta}} < (1 + 8\epsilon) e^{-8\epsilon}.$$

When $\epsilon \in (0, \frac{1}{4})$, $(1 + 8\epsilon) e^{-8\epsilon}$ is strictly less than 1. Therefore, we have:

$$\left[ (1 + 8\epsilon) \left( \frac{1 - (1 + 8\epsilon)\theta}{1 - \theta} \right)^{\frac{1-\theta}{\theta}} \right]^{\theta T_j - \Delta}$$

$$\geq \left[ (1 + 8\epsilon) \left( \frac{1 - (1 + 8\epsilon)\theta}{1 - \theta} \right)^{\frac{1-\theta}{\theta}} \right]^{\theta T}$$

$$\geq \left[ (1 + 8\epsilon)(1 - 8\epsilon) \right]^{\theta T}$$

$$= \left( 1 - 64\epsilon^2 \right)^{\frac{1}{2304\epsilon^2} \ln \frac{k}{\delta}}$$

$$\geq (\delta/k)^{1/18}.$$

Combining the above with (15), we have $\frac{\Pr_{I \cup \{a_j\}}[W]}{\Pr_I[W]} \geq (\delta/k)^{5/9} \geq 5\delta/k$ for any $\delta \in (0, \frac{1}{48})$.

$\square$

We will now proceed to analyze the probability of $\mathcal{E}_B$ when $\mathcal{A}$ is executed on $\Theta_{I \cup \{a_j\}}$. For this purpose, define $1_S(W)$ to be an indicator function that equals 1 if the history $W$ results in the occurrence of $\mathcal{E}_S$, and 0 otherwise (note that once $W$ is given, the output of $\mathcal{A}$ is fully determined, and hence, so is $\mathcal{E}_S$). With this, we can derive:

$$\Pr_{I \cup \{a_j\}}[\mathcal{E}_B] \geq \Pr_{I \cup \{a_j\}}[\mathcal{E}_S]$$

$$= \sum_{W \text{ s.t. } 1_S(W) = 1} \Pr_{I \cup \{a_j\}}[W]$$

$$(\text{by Lemma 8}) \geq \sum_{W \text{ s.t. } 1_S(W) = 1} \Pr_I[W] \cdot \frac{5\delta}{k}$$

$$\geq (5\delta/k) \Pr_I[\mathcal{E}_S]$$

$$(\text{by (13)}) > 2\delta/k.$$

Therefore, we have proved that for any $I \in \mathcal{I}_{k-1}$ and any ordinary arm $a_j \in U \setminus I$, the failure probability of $\mathcal{A}$ under instance $\Theta_{I \cup \{a_j\}}$ exceeds $\frac{2\delta}{k}$.

Recall that for any $I \in \mathcal{I}_{k-1}$, there are at least $\frac{n-k}{2}$ ordinary arms $a_j$ in $U \setminus I$. Summing up the failure probabilities of $\mathcal{A}$ on the instances $\Theta_J$ of all $J \in \mathcal{I}_k$ leads to:

$$\sum_{J \in \mathcal{I}_k} \Pr_J[V \neq J]$$

$$\geq \sum_{J \in \mathcal{I}_k} \sum_{a_j \in J} \Pr_J[V = J \cup \{a_1\} \setminus \{a_j\}]$$

$$= \sum_{I \in \mathcal{I}_{k-1}} \sum_{a_j \in U \setminus I} \sum_{\substack{J \in \mathcal{I}_k \text{ s.t.} \\ I \cup \{a_j\} = J}} \Pr_J[V = I \cup \{a_1\}]$$

$$= \sum_{I \in \mathcal{I}_{k-1}} \sum_{a_j \in U \setminus I} \Pr_{I \cup \{a_j\}}[V = I \cup \{a_1\}]$$

$$= \sum_{I \in \mathcal{I}_{k-1}} \sum_{a_j \in U \setminus I} \Pr_{I \cup \{a_j\}}[\mathcal{E}_B]$$

$$> \sum_{I \in \mathcal{I}_{k-1}} \frac{n - k}{2} \cdot \frac{2\delta}{k}$$

$$= \binom{n-1}{k-1} \frac{n-k}{k} \delta = \binom{n-1}{k} \delta = |\mathcal{I}_k| \delta.$$

By the pigeon hole principle, there is at least one $J \in \mathcal{I}_K$ such that $\Pr_J[V \neq J] > \delta$. As this is not allowed, we conclude that $\mathcal{A}$ must take at least $\frac{n}{36864\epsilon^2} \frac{1}{\theta} \ln \frac{k}{\delta}$ samples on the $\Theta_I$ of at least one $I \in \mathcal{I}_{k-1}$.

**Randomized Algorithms.** It is standard to extend the above deterministic lower bound to randomized algorithms—such an argument can be found, for example, in [16]. With this, we complete the whole proof of Theorem 2.

## C Analysis of Top-$k_{\mathsf{add}}$ Arm Selection

Recall that we define $k_{\mathsf{add}}$-AS by changing the multiplicative guarantee $\theta_i(V) \geq (1 - \epsilon)\theta_i(B)$ in $k$-AS to an additive guarantee $\theta_i(V) \geq \theta_i(B) - \epsilon'$.

The algorithm of $k_{\mathsf{add}}$-AS is very similar to $k$-AS. As we do not require a multiplicative guarantee here, we simply drop the "guess" stage and adjust the parameters appropriately, as shown in Algorithm 7, 8. The sample complexity of AME is exactly the same as the algorithm in [10] for EXPLORE-$k$ as shown in Theorem 3. However our problem guarantees that $\theta_i(V) \geq \theta_i(B) - \epsilon'$ for all $i \in [k]$ while EXPLORE-$k$ guarantees that $\theta_i(V) \geq \theta_k(B) - \epsilon'$ for all $i \in [k]$ as we discussed in Section 1.2. Thus, $k_{\mathsf{add}}$-AS achieves a strictly stronger guarantee than EXPLORE-$k$. For the lower bound, as the $k_{\mathsf{add}}$-AS problem is stronger than EXPLORE-$k$, the lower bound for EXPLORE-$k$ directly applies to $k_{\mathsf{add}}$-AS.

---

**Algorithm 7:** Additive Median Elimination (AME)

---

**1** **input:** $B, \epsilon, \delta, k$
**2** $S_1 = B, \epsilon_1 = \epsilon/16, \delta_1 = \delta/8$ and $\ell = 1$;
**3** **while** $|S_\ell| > 4k$ **do**
**4** $\quad$ sample every arm $a \in S_\ell$ for $Q_\ell = (12/\epsilon_\ell^2) \log(6k/\delta_\ell)$ times;
**5** $\quad$ **for** *each arm* $a \in S_\ell$ **do**
**6** $\quad\quad$ its empirical value $\hat{\theta}(a)$ = the average of the $Q_\ell$ samples from $a$;
**7** $\quad$ $a_1, \ldots, a_{|S_\ell|}$ = the arms sorted in non-increasing order of their empirical values;
**8** $\quad$ $S_{\ell+1} = \{a_1, \ldots, a_{|S_\ell|/2}\}$;
**9** $\quad$ $\epsilon_{\ell+1} = 3\epsilon_\ell/4, \delta_{\ell+1} = \delta_\ell/2$ and $\ell = \ell + 1$;
**10** **return** $S_\ell$;

---

**Algorithm 8:** Additive Uniform Sampling (AUS)

---

**1** **input:** $S, \epsilon, \delta, k$
**2** sample every arm $a \in S$ for $Q = (96/\epsilon^2) \log(4|S|/\delta)$ times;
**3** **for** *each arm* $a \in S$ **do**
**4** $\quad$ its US-empirical value $\hat{\theta}^{US}(a)$ = the average of the $Q$ samples from $a$;
**5** $a_1, \ldots, a_{|S|}$ = the arms sorted in non-increasing order of their US-empirical values;
**6** **return** $\{(a_1, \hat{\theta}^{US}(a_1)), \ldots, (a_k, \hat{\theta}^{US}(a_k))\}$

---

**Theorem 3** *AME solves the $k_{\mathsf{add}}$-AS problem with expected cost $O\left(\frac{n}{\epsilon^2} \log \frac{k}{\delta}\right)$.*

*Proof*: The proof the Theorem 3 directly follows Lemma 1 to Lemma 4, except that we change the multiplicative guarantee into the additive guarantee and utilize the Chernoff bound (6) in the analysis instead. $\qquad \square$

## D Analysis of $k$-MOST CONNECTED VERTEX

This section is devoted to the proof of Theorem 4:

**Theorem 4** *For any $\epsilon \in \left(0, \frac{1}{12}\right)$ and $\delta \in \left(0, \frac{1}{48}\right)$, the following statements are true about any $k$-MCV algorithm:*

- *when $\deg_k(B) \geq \Omega\left(\frac{1}{\epsilon^2} \log \frac{n}{\delta}\right)$, there is an instance on which the algorithm must probe $\Omega(\frac{n}{\epsilon^2} \frac{m}{\deg_k(B)} \log \frac{k}{\delta})$ edges in expectation.*

- *when $\deg_k(B) < O(\frac{1}{\epsilon})$, there is an instance on which the algorithm must probe $\Omega(nm)$ edges in expectation.*

**Proof of Theorem 4 for Large $\deg_k(B)$.** We prove the first branch of the theorem by a reduction from $k$-AS. Let $\mathcal{A}$ be an algorithm solving $k$-MCV with expected cost $Q$. We will deploy $\mathcal{A}$ to solve the hard $k$-AS instances created earlier in the proof of Theorem 2 with sample complexity $Q$, and thereby obtaining a lower bound for $Q$. Recall that, to prove Theorem 2 in Appendix B.3, we associated each $I \in \mathcal{I}_{k-1}$ with an instance $\Theta_I$ for $k$-AS that is described by (9), (10), and (11). From $\Theta_I$, we construct a random hidden bipartite graph $G = (B, W, E)$ with $|B| = n$ and $|W| = m$ where $m \geq \frac{24}{\epsilon^2} \frac{1}{\theta} \log \frac{n}{\delta}$. For the $i$-th black vertex, we independently make each edge incident on it solid with probability $\theta(a_i)$. The graph $G$ thus constructed is fed as an $k$-MCV input to $\mathcal{A}$. Finally, we select the arm $a_i$ into the solution of $k$-AS if and only if $\mathcal{A}$ includes the $i$-th black vertex in its ($k$-MCV) solution.

We denote by $d_i$ the degree of the $i$-th black vertex in $G$; notice that $d_i$ is a random variable. Set $t = \theta m$. By Chernoff bound, we know:

$$
\begin{aligned}
& \Pr\left[d_1 \leq (1 - \epsilon/2)(1 + 4\epsilon)\theta m\right] \\
= \ & \Pr\left[d_1 \leq (1 - \epsilon/2)(1 + 4\epsilon)t\right] \\
\leq \ & \exp\left(-\frac{(\epsilon/2)^2}{2}(1 + 4\epsilon)t\right).
\end{aligned}
$$

Therefore, if $t \geq \frac{24}{\epsilon^2} \log \frac{n}{\delta}$ and $\epsilon \in (0, \frac{1}{12})$, with probability at least $1 - \frac{\delta}{n}$, we have $d_1 \geq (1 - \frac{\epsilon}{2})(1 + 4\epsilon)t \geq (1 + 3\epsilon)t$. Similarly, we can show that with probability at least $1 - \frac{\delta}{n}$, we have $d_i \geq (1 + 7\epsilon)t$ for any $a_i \in I$ and $d_i \leq (1 + \epsilon)t$ for any $a_i \in U \setminus I$. Notice that both $(1 - \epsilon)(1 + 7\epsilon)t > (1 + \epsilon)t$ and $(1 - \epsilon)(1 + 3\epsilon)t > (1 + \epsilon)t$ hold. By the union bound, with probability at least $1 - \delta$, $\mathcal{A}$ must select the first vertex and the vertices corresponding to the arms of $I$ as its ($k$-MCV) solution set. Thus, we have obtained an $(\epsilon, 2\delta)$-approximate algorithm that solves $k$-AS on instance $\Theta_I$.

On the other hand, the proof of Theorem 2 has showed that any $(\epsilon, 2\delta)$-approximate $k$-AS algorithm must take in expectation $\Omega(\frac{n}{\epsilon^2} \frac{1}{\theta_k(B)} \log \frac{k}{2\delta})$ samples on the $\Theta_I$ of at least one $I \in \mathcal{I}_{k-1}$. It thus follows that $Q = \Omega(\frac{n}{\epsilon^2} \frac{m}{\deg_k(B)} \log \frac{k}{\delta})$.

**Proof of Theorem 4 for Small $\deg_k(B)$.** Let us define the *single-vertex* problem:

> Suppose that $t < \frac{1-2\epsilon}{2\epsilon}$ is a given positive integer. The bipartite graph only has one black vertex $b$, and $m \geq \max\{t+1, 64\}$ white vertices. The goal is to distinguish whether $\deg(b)$ is $t$ or $t + 1$ with probability at least $\frac{15}{16}$.

We now give a lower bound for the above problem.

**Lemma 9** *Any algorithm solving the above single-vertex problem must issue $\Omega(m)$ probe operations.*

*Proof*: Our proof is based on a sampling lower bound established in [3] for evaluating symmetric functions[2]. It is showed in [3] that any sampling algorithm for symmetric functions can be simulated by an i.i.d. sampling algorithm under some specific conditions. Therefore, the sampling lower bound can be measured by the distance between two i.i.d. distributions. Formally, we need the following definition:

**Definition 1** [Total Variational Distance] *Suppose $X$ and $Y$ are two discrete distributions with a common support $\mathcal{S}$. The* total variational distance *(also known as statistical distance) between $X$ and $Y$ is defined as*

$$
d_{\mathsf{TV}}(X, Y) = \frac{1}{2} \sum_{s \in \mathcal{S}} |\Pr[X = s] - \Pr[Y = s]|.
$$

Suppose $f(x_1, \ldots, x_m) : \mathcal{S}^m \to \mathcal{Z}$ is a symmetric function where $\mathcal{S}$ and $\mathcal{Z}$ are arbitrary sets. For any $x \in \mathcal{S}^m$, we use $U_x$ to denote the distribution of i.i.d. query outcomes on $x$ supported on $\mathcal{X}$, i.e., $U_x(s) = \frac{1}{m} \sum_{i=1}^{m} 1(x_i = s), \forall s \in \mathcal{S}$ where $1()$ is an indicator function.

**Lemma 10** ([3, Theorem 4.21]) *For any $\epsilon \in (0,1), \delta \in (\frac{2}{m}, \frac{1}{8})$, given two inputs $x, y \in \mathcal{S}^m$, if a sampling algorithm $\mathcal{A}$ can distinguish whether $f = f(x)$ or $f = f(y)$ with probability at least $1 - \delta$ within worst case sampling complexity at most $\frac{m}{2} - \sqrt{m/(6\delta)}$, it must take at least*

$$\frac{1}{8 \cdot d_{\mathsf{TV}}(U_x, U_y)} \ln \frac{1}{8\delta}$$

*samples in the worst case.*

Consider the simple symmetric function $\deg(b) = f(x_1, ..., x_m) = \sum_{i=1}^m x_i$ where $x_i$ is 1 if the edge connecting $b$ and the $i$-th white vertex is solid, and 0 otherwise. We use $x$ to denote an input with $f = t$. Then, we have that $U_x(0) = \frac{m-t}{m}$ and $U_x(1) = \frac{t}{m}$. Similarly, we use $y$ to denote an input with $f = t + 1$ and we have that $U_y(0) = \frac{m-t-1}{m}$ and $U_y(1) = \frac{t+1}{m}$.

By the definition of $d_{\mathsf{TV}}$, it is easy to see that $d_{\mathsf{TV}}(U_x, U_y) = \frac{1}{m}$. Therefore, we conclude that the lower bound of the single vertex problem is $\Omega(m)$.

$\square$

The lemma below shows a reduction from the above problem to $k$-MCV.

**Lemma 11** *For any $\epsilon \in (0, \frac{1}{12}), \delta \in (0, \frac{1}{48})$, given an $(\epsilon, \delta)$-approximate $k$-MCV algorithm $\mathcal{A}$ with expected sample complexity $Q$, we can design an algorithm $\mathcal{B}$ to solve the single-vertex problem with at most $32Q/n$ probe operations.*

*Proof*: We reduce the single-vertex problem to 1-MCV as follows. First, pick a value $n \geq 32$ and construct an edge hidden graph $G$ with $n$ black vertices and $m \geq \max\{t+1, 64\}$ white vertices. Denote the set of black vertices as $\{b_1, \ldots, b_n\}$. Randomly choose a black vertex $b_i$ as the *pivot vertex*. For any $b_j$ with $j \neq i$, we connect it to $t$ white vertices randomly chosen without replacement (i.e., pick a random $t$-sized subset of white vertices over all the $\binom{m}{t}$ possible choices).

Algorithm $\mathcal{B}$ solves the single-vertex problem by simulating algorithm $\mathcal{A}$ in solving the 1-MCV problem on $G$. Whenever $\mathcal{A}$ probes a hidden edge of $b_j$ with $j \neq i$, $\mathcal{B}$ simply does the same. On the other hand, when $\mathcal{A}$ probes a hidden edge of the pivot vertex $b_i$, $\mathcal{B}$ probes a hidden edge of the vertex $b$ in the single-vertex problem, and passes the result to $\mathcal{A}$. If (i) the pivot vertex is probed less than $\frac{32Q}{n}$ times and at the same time (ii) $\mathcal{A}$ returns a vertex $b_j$ with $j \neq i$, $\mathcal{B}$ decides $\deg(b) = t$. In all other cases, $\mathcal{B}$ decides $\deg(b) = t + 1$.

This finishes the description of $\mathcal{B}$. Its worst case cost is clearly $\frac{32Q}{n}$ (for the single-vertex problem). Next, we show that $\mathcal{B}$ decides $\deg(b)$ correctly with probability at least $\frac{15}{16}$.

Consider first the case of $\deg(b) = t$. All the black vertices in $G$ have degrees exactly $t$. Denote by $T_b$ the number of probe operations that $\mathcal{B}$ performs on $b$. As the pivot vertex is indistinguishable from any other verties, we have $\mathbb{E}[T_b] = \frac{Q}{n}$. Markov's inequality then gives:

$$\Pr\left[T_b \geq \frac{32Q}{n}\right] \leq \frac{1}{32}.$$

If $T_b < \frac{32Q}{n}$, $\mathcal{B}$ errs if and only if $\mathcal{A}$ returns $b_i$. Since $b$ is indistinguishable, the probability that $\mathcal{A}$ returns $b_i$ is $1/n$ (recall that every black vertex has the same chance of being the pivot vertex). Therefore, with probability at least $1 - \frac{1}{n} - \frac{1}{32} \geq \frac{15}{16}$, $\mathcal{B}$ returns a correct answer.

Next, consider the case of $\deg(b) = t + 1$. $\mathcal{B}$ makes a mistake only if $T_b < \frac{32Q}{n}$ and $\mathcal{A}$ returns a vertex $b_j$ with $j \neq i$. Since $\mathcal{A}$ is an $(\epsilon, \delta)$-approximate 1-MCV algorithm, the probability that $\mathcal{B}$ errs is at most $\delta \leq \frac{1}{16}$. Hence, once again, it is correct with probability at least $\frac{15}{16}$. $\square$

The second branch of Theorem 4 follows from Lemmas 9 and 11.

# E  Analysis of Top-$k_{\mathsf{avg}}$ Arm Selection

In this section we analyze our algorithm for $k_{\mathsf{avg}}$-AS as shown in 4, 5, 6. We first present the analysis of Quartile Elimination in Appendix E.1. Next, we prove Theorem 5 in Appendix E.2. Finally, we prove Theorem 6 in Appendix E.3.

### E.1 Analysis of Quartile Elimination

This subsection serves as a proof for the following lemma:

**Lemma 12** *In Quartile Elimination, if $\mu \leq \theta_{\mathsf{avg}}(B)$, with probability at least $1 - \frac{\delta}{4}$, QE returns an arm set $S$ satisfying $\theta_{\mathsf{avg}}(S) \geq (1 - \frac{\epsilon}{2})\theta_{\mathsf{avg}}(B)$.*

Focus on the $\ell$-th round for a specific $\ell$. For any subset $X \subseteq S_\ell$, let $\hat{\theta}_{\mathsf{avg}}(X) = \frac{1}{k}\sum_{i=1}^k \hat{\theta}_i(X)$. We use $m_\ell$ to denote the median of the means of all the arms in $S_\ell$, and $\tau_\ell$ to denote the largest empirical value of the eliminated arms in this round. For each $i \leq k$, define a Boolean variable $X_{\ell,i}$ which equals 1 if $\hat{\theta}_i(S_\ell) < m_\ell + \frac{\epsilon_\ell \theta_{\mathsf{avg}}(S_\ell)}{2}$, and 0 otherwise. Let $X_\ell = \frac{1}{k}\sum_{i=1}^k (\theta_i(S_\ell) - m_\ell) X_{\ell,i}$. Furthermore, define $\mathcal{E}_\ell^A$ to be the event that $X_\ell \leq \epsilon_\ell \theta_{\mathsf{avg}}(S_\ell)$ holds, and $\mathcal{E}_\ell^B$ to be the event that $\tau_\ell < m_\ell + \frac{\epsilon_\ell \theta_{\mathsf{avg}}(S_\ell)}{2}$ holds.

**Lemma 13** *In Round $\ell$, if both $\mathcal{E}_\ell^A$ and $\mathcal{E}_\ell^B$ occur, then $\theta_{\mathsf{avg}}(S_{\ell+1}) \geq (1 - \epsilon_\ell)\theta_{\mathsf{avg}}(S_\ell)$ holds.*

*Proof*: First, we claim that, under $\mathcal{E}_\ell^A$ and $\mathcal{E}_\ell^B$, it holds that

$$\frac{1}{k}\sum_{i=1}^k \theta_i(S_{\ell+1}) \geq \frac{1}{k}\sum_{i=1}^k \left((1 - X_{\ell,i})\theta_i(S_\ell) + X_{\ell,i}m_\ell\right). \tag{17}$$

To see this, consider an arm $a_i(S_\ell)$ with $i \leq k$. If $X_{\ell,i} = 0$, by definition, we have $\hat{\theta}_i(S_\ell) \geq m_\ell + \frac{\epsilon_\ell \theta_{\mathsf{avg}}(S_\ell)}{2}$. Since $\mathcal{E}_\ell^B$ occurs, we have that

$$\hat{\theta}_i(S_\ell) \geq m_\ell + \frac{\epsilon_\ell \theta_{\mathsf{avg}}(S_\ell)}{2} > \tau_\ell.$$

Therefore, we know that $a_i(S_\ell)$ has not been eliminated in round $\ell$. Since $a_i(S_\ell)$ is one of the top-$k$ arms in $S_\ell$, it must be one of the top-$k$ arms in $S_{\ell+1}$. Hence, for each arm $a_i(S_\ell)$ with $i \leq k$ and $X_{\ell,i} = 0$, $\theta_i(S_\ell)$ appears in both LHS and RHS of (17) exactly once. Notice that in QE, $k \leq \frac{|S_\ell|}{4}$. After eliminating $\frac{|S_\ell|}{4}$ arms, there exist at least $k$ arms with means at least $m_\ell$. Thus, each term in LHS of (17) is at least $m_\ell$. Therefore, we conclude that our claim (17) is true. Now, we can see that:

$$
\begin{aligned}
\theta_{\mathsf{avg}}(S_{\ell+1}) &\geq \frac{1}{k}\sum_{i=1}^k \left((1 - X_{\ell,i})\theta_i(S_\ell) + X_{\ell,i}m_\ell\right) \\
&= \theta_{\mathsf{avg}}(S_\ell) - X_\ell \\
\text{(by } \mathcal{E}_\ell^A) \quad &\geq (1 - \epsilon_\ell)\theta_{\mathsf{avg}}(S_\ell).
\end{aligned}
$$

$\square$

**Lemma 14** *Under the condition $\mu \leq \theta_{\mathsf{avg}}(B)$, if for all $j < \ell$, Quartile Elimination ensures*
$$\theta_{\mathsf{avg}}(S_{j+1}) \geq (1 - \epsilon_j)\theta_{\mathsf{avg}}(S_j) \tag{18}$$
*then, with probability at least $1 - \delta_\ell$, $\theta_{\mathsf{avg}}(S_{\ell+1}) \geq (1 - \epsilon_\ell)\theta_{\mathsf{avg}}(S_\ell)$ holds.*

*Proof*: By Lemma 13, it suffices to prove that with probability no less than $1 - \delta_\ell$, both $\mathcal{E}_\ell^A$ and $\mathcal{E}_\ell^B$ occur.

First we show that $\Pr[\mathcal{E}_\ell^A] = \Pr[X_\ell \leq \epsilon_\ell \theta_{\mathsf{avg}}(S_\ell)] \geq 1 - \frac{\delta_\ell}{2}$. From (18), $\mu_\ell \leq \prod_{j=1}^{\ell-1}(1 - \epsilon_j)\theta_{\mathsf{avg}}(S_1) \leq \theta_{\mathsf{avg}}(S_\ell)$. For any $i \leq k$, define $\beta_{\ell,i} = \max\{0, \theta_i(S_\ell) - m_\ell - \frac{\epsilon_\ell \theta_{\mathsf{avg}}(S_\ell)}{2}\}$ and $Y_{\ell,i} = \beta_{\ell,i}X_{\ell,i}$. We have:

$$
\begin{aligned}
X_\ell &\leq \frac{1}{k}\sum_{i=1}^k Y_{\ell,i} + \frac{1}{k}\sum_{i=1}^k \frac{\epsilon_\ell \theta_{\mathsf{avg}}(S_\ell)}{2}X_{\ell,i} \\
&\leq \frac{1}{k}\sum_{i=1}^k Y_{\ell,i} + \frac{\epsilon_\ell \theta_{\mathsf{avg}}(S_\ell)}{2}.
\end{aligned}
$$

Hence, it suffices to show $\Pr[\mathcal{E}_\ell^A] \leq \Pr[\frac{1}{k} \sum_{i=1}^k Y_{\ell,i} \leq \frac{\epsilon_\ell \theta_{\text{avg}}(S_\ell)}{2}] \leq \frac{\delta_\ell}{2}$. Let $Y_\ell = \sum_{i=1}^k Y_{\ell,i}$. We have:

$$\Pr\left[ \frac{1}{k} \sum_{i=1}^k Y_{\ell,i} \geq \epsilon_\ell \theta_{\text{avg}}(S_\ell)/2 \right]$$

$$= \Pr\left[ \sum_{i=1}^k \epsilon_\ell Q_\ell Y_{\ell,i}/2 \geq \epsilon_\ell^2 Q_\ell \theta_{\text{avg}}(S_\ell)k/4 \right]$$

$$= \Pr\left[ \exp\left( \sum_{i=1}^k \epsilon_\ell Q_\ell Y_{\ell,i}/2 \right) \geq \exp\left( \epsilon_\ell^2 Q_\ell \theta_{\text{avg}}(S_\ell)k/4 \right) \right]$$

$$\leq \frac{\mathbb{E}\left[ \exp\left( \epsilon_\ell Q_\ell Y_\ell/2 \right) \right]}{\exp\left( \epsilon_\ell^2 Q_\ell \theta_{\text{avg}}(S_\ell)k/4 \right)}. \tag{19}$$

For all $a_i(S_\ell)$ satisfying $\theta_i(S_\ell) - m_\ell - \frac{\epsilon \theta_{\text{avg}}(S_\ell)}{2} > 0$ where $i \leq k$, we use $U$ to denote the set of their indexes and $\mathcal{I}$ to denote the collection of all non-empty subsets of $U$. For any $I \in \mathcal{I}$, denote by $\mathcal{E}^I$ the event that both of the following are true: (i) $X_{\ell,i} = 1$ holds for all $i \in I$, and (ii) $X_{\ell,i} = 0$ holds for all $i \in U \backslash I$. We have:

$$\Pr\left[ \mathcal{E}^I \right] \leq \Pr\left[ \bigwedge_{i \in I}(X_{\ell,i} = 1) \right]$$

$$\leq \Pr\left[ \sum_{i \in I} \hat{\theta}_i(S_\ell) < |I| \left( m_\ell + \frac{\epsilon_\ell \theta_{\text{avg}}(S_\ell)}{2} \right) \right]$$

$$= \Pr\left[ \sum_{i \in I} \hat{\theta}_i(S_\ell) < \sum_{i \in I} \theta_i(S_\ell) - \sum_{i \in I} \beta_{\ell,i} \right]$$

$$\leq \exp\left( -\frac{\left( \sum_{i \in I} \beta_{\ell,i} \right)^2}{2} \frac{Q_\ell}{\sum_{i \in I} \theta_i(S_\ell)} \right). \tag{20}$$

where the last inequality is due to Chernoff bound (5).

Let $\eta_I = \frac{\sum_{i \in I} \beta_{\ell,i}}{\sum_{i \in I} \theta_i(S_\ell)}$. By (20), we have:

$$\mathbb{E}\left[ \exp\left( \epsilon_\ell Q_\ell Y_\ell/2 \right) \right]$$

$$\leq \sum_{I \in \mathcal{I}} \exp\left( \epsilon_\ell Q_\ell \sum_{i \in I} \beta_{\ell,i}/2 \right) \Pr\left[ \mathcal{E}^I \right] + \Pr\left[ Y_\ell = 0 \right]$$

$$\leq \sum_{I \in \mathcal{I}} \exp\left( \eta_I \left( \frac{\epsilon_\ell}{2} - \frac{1}{2}\eta_I \right) Q_\ell \sum_{i \in I} \theta_i(S_\ell) \right) + 1$$

$$\leq 2^k \exp\left( \frac{\epsilon_\ell^2}{8} Q_\ell \theta_{\text{avg}}(S_\ell)k \right).$$

Therefore, we get:

$$\Pr\left[ \frac{1}{k} \sum_{i=1}^k Y_{\ell,i} \geq \frac{\epsilon_\ell \theta_{\text{avg}}(S_\ell)}{2} \right]$$

$$\leq 2^k \exp\left( -\frac{\epsilon_\ell^2}{8} Q_\ell \theta_{\text{avg}}(S_\ell)k \right)$$

$$\leq \left( \frac{2}{e} \right)^k \frac{\delta_\ell}{2} \leq \frac{\delta_\ell}{2}.$$

Next, we prove that $\Pr[\mathcal{E}_\ell^B] = \Pr[\tau_\ell < m_\ell + \frac{\epsilon_\ell \theta_{\text{avg}}(S_\ell)}{2}] \geq 1 - \frac{\delta_\ell}{2}$. For any $i \leq |S_\ell|$, define a Boolean variable $Z_i$ that equals 1 if $\hat{\theta}_i(S_\ell) \geq m_\ell + \frac{\epsilon_\ell \theta_{\text{avg}}(S_\ell)}{2}$, or 0 otherwise. It suffices to show that at least

$\frac{|S_\ell|}{4}$ arms have empirical values smaller than $m_\ell + \frac{\epsilon_\ell \theta_{\text{avg}}(S_\ell)}{2}$, i.e., $\sum_{i=1}^{|S_\ell|} Z_i \leq \frac{3|S_\ell|}{4}$. We show that with a large probability $\sum_{i=|S_\ell|/2}^{|S_\ell|} Z_i \leq \frac{|S_\ell|}{4}$ holds. By Chernoff bounds, for any $|S_\ell|/2 \leq i \leq |S_\ell|$, we have:

$$
\begin{aligned}
\mathbb{E}\left[Z_i\right] &= \Pr\left[\hat{\theta}_i(S_\ell) \geq m_\ell + \frac{\epsilon_\ell \theta_{\text{avg}}(S_\ell)}{2}\right] \\
&\leq \Pr\left[\hat{\theta}_i(S_\ell) \geq \theta_i(S_\ell) + \frac{\epsilon_\ell \theta_{\text{avg}}(S_\ell)}{2}\right] \\
&\leq \exp\left(-\frac{\epsilon_\ell^2}{12} Q_\ell \theta_{\text{avg}}(S_\ell)\right).
\end{aligned}
$$

Let $\lambda = \exp\left(-\frac{\epsilon_\ell^2}{12} Q_\ell \theta_{\text{avg}}(S_\ell)\right)$. It is easy to verify that $\lambda < \frac{1}{2}$. By Chernoff bound (7), we have

$$
\begin{aligned}
\Pr\Bigg[\sum_{i=|S_\ell|/2}^{|S_\ell|} Z_i &\geq \frac{|S_\ell|}{4}\Bigg] \\
&\leq \left(\left(\frac{\lambda}{1/2}\right)^{1/2}\left(\frac{1-\lambda}{1/2}\right)^{1/2}\right)^{|S_\ell|/2} \\
&\leq \left(\sqrt{2\lambda}\cdot\sqrt{2}\right)^{|S_\ell|/2} \\
&\leq \exp\left(\frac{|S_\ell|}{2}\left(\ln(2) - \frac{\epsilon_\ell^2}{24} Q_\ell \theta_{\text{avg}}(S_\ell)\right)\right) \\
&\leq \exp\left(-\frac{|S_\ell|}{2}\frac{\epsilon_\ell^2}{48} Q_\ell \theta_{\text{avg}}(S_\ell)\right) \\
&\leq \exp\left(-\frac{\epsilon_\ell^2}{24} Q_\ell k \theta_{\text{avg}}(S_\ell)\right) \leq \frac{\delta_\ell}{2},
\end{aligned}
$$

where the third to last inequality used the fact that $\ln(2) - \frac{\epsilon_\ell^2}{24} Q_\ell \theta_{\text{avg}}(S_\ell) \leq -\frac{\epsilon_\ell^2}{48} Q_\ell \theta_{\text{avg}}(S_\ell)$ by our setting of $Q_\ell$ and the second to last inequality used the fact that $|S_\ell| \geq 4k$.

$\square$

Now, suppose QE terminates in $L$ rounds. By summing up the failure probabilities and the errors of all rounds in QE, we have that $\sum_{i=1}^{L} \delta_i \leq \delta$ and $\prod_{i=1}^{L}(1 - \epsilon_i) \geq 1 - \epsilon/2$. Thus, we complete the proof of Lemma 12.

### E.2  Correctness and Sample Complexity

**Analysis of Uniform Sampling.** Let us start with:

**Lemma 15** *In Uniform Sampling, if $\mu_s \leq \theta_{\text{avg}}(S)$, then with probability at least $1 - \frac{\delta}{4}$, US returns an arm set $V$ which satisfies $\theta_{\text{avg}}(V) \geq (1 - \frac{\epsilon}{2})\theta_{\text{avg}}(S)$.*

*Proof*: We use $\mathcal{U}$ to denote the collection of all $k$-sized subsets of $S$. Consider the subset $U^* \in \mathcal{U}$ with the largest average mean. By definition we have $\theta_{\text{avg}}(S) = \theta_{\text{avg}}(U^*)$. By Chernoff bound (2),

$$
\begin{aligned}
\Pr\left[\hat{\theta}_{\text{avg}}(U^*) \leq \left(1 - \frac{\epsilon}{4}\right)\theta_{\text{avg}}(U^*)\right] \\
\leq \exp\left(-\frac{\epsilon^2}{48} Q \theta_{\text{avg}}(S)k\right).
\end{aligned}
$$

Consider an arbitrary set $U \in \mathcal{U}$. Let $\alpha = \frac{\theta_{\mathsf{avg}}(U^*)}{\theta_{\mathsf{avg}}(U)}\left(1 - \frac{\epsilon}{4}\right) - 1$. If $\theta_{\mathsf{avg}}(U) < (1 - \epsilon/2)\theta_{\mathsf{avg}}(U^*)$ but $\hat{\theta}_{\mathsf{avg}}(U) \geq (1 - \epsilon/4)\theta_{\mathsf{avg}}(U^*)$, we obtain:

$$
\begin{aligned}
&\Pr\left[\hat{\theta}_{\mathsf{avg}}(U) \geq \left(1 - \frac{\epsilon}{4}\right)\theta_{\mathsf{avg}}(U^*)\right] \\
=\ &\Pr\left[\hat{\theta}_{\mathsf{avg}}(U) \geq (1 + \alpha)\theta_{\mathsf{avg}}(U)\right] \\
\leq\ &\exp\left(-\frac{\epsilon^2}{48}Q\theta_{\mathsf{avg}}(S)k\right),
\end{aligned}
$$

where the last inequality is due to Chernoff bounds (1) and (3) (we distinguish $0 < \alpha < 1$ and $\alpha \geq 1$. The calculation details are similar to the proof of Lemma 3.

Notice that in US we have $|S| \leq 4k$. Applying the union bound over all the subsets of size $k$, we assert that the failure probability of US is at most

$$
\binom{|S|}{k}\exp\left(-\frac{\epsilon^2}{48}Q\theta_{\mathsf{avg}}(S)k\right) \quad \leq \quad \frac{\delta}{4}.
$$

$\qquad\qquad\qquad\qquad\qquad\qquad\qquad\qquad\qquad\qquad\qquad\qquad\qquad\quad\square$

Next, we show that with high probability QE-AS terminates at $\mu \in [\frac{1}{8}\theta_{\mathsf{avg}}(B), \theta_{\mathsf{avg}}(B)]$. If this is not what happens, one of the following two events must have happened:

(1) *Premature*: QE-AS terminates when $\mu > \theta_{\mathsf{avg}}(B)$.

(2) *Overdue*: QE-AS does not terminate after invoking US with $\mu < \frac{1}{4}\theta_{\mathsf{avg}}(B)$.

We say that QE *succeeds* if the arm set $S$ it returns satisfies $\theta_{\mathsf{avg}}(S) \geq (1 - \frac{\epsilon}{2})\theta_{\mathsf{avg}}(B)$.

**Lemma 16** *Both of the following statements are true:*

- *The premature event happens with probability at most $\delta/4$.*

- *If QE succeeds, the overdue event happens with probability at most $\delta/4$.*

*Proof*: The second statement directly follows from the proof of Lemma 5. For the first lemma, let us focus on a specific guess $\mu$. Let $\nu = \frac{\mu}{\theta_{\mathsf{avg}}(B)}$. We use $\mathcal{U}$ to denote the collection of all $k$-sized subsets in $S$. Consider a specific $U \in \mathcal{U}$. Let $\alpha = 2\frac{\mu}{\theta_{\mathsf{avg}}(U)} - 1$. We claim:

$$
\begin{aligned}
&\Pr[\hat{\theta}_{\mathsf{avg}}(U) \geq 2\mu] \\
=\ &\Pr[\hat{\theta}_{\mathsf{avg}}(U) \geq (1 + \alpha)\theta_{\mathsf{avg}}(U)] \\
\leq\ &\frac{\delta/4}{2\nu}\exp\left(-\frac{5}{2}k\right).
\end{aligned}
\tag{21}
$$

Notice that here we have $\alpha > 1$. By Chernoff bound (3),

$$
\begin{aligned}
&\Pr[\hat{\theta}_{\mathsf{avg}}(U) \geq (1 + \alpha)\theta_{\mathsf{avg}}(U)] \\
\leq\ &\exp\left(-2Q\mu k/6\right) \leq \frac{\delta/4}{4}\exp\left(-\frac{5}{2}k\right).
\end{aligned}
$$

This proves (22) for $1 \leq \nu < 2$.

When $\nu \geq 2$, by Chernoff bound (4),

$$
\begin{aligned}
&\Pr[\hat{\theta}_{\mathsf{avg}}(U) \geq (1 + \alpha)\theta_{\mathsf{avg}}(U)] \\
\leq\ &\left(\frac{e}{2\mu/\theta_{\mathsf{avg}}(U)}\right)^{2Q\mu k} \\
\leq\ &\left(\frac{e}{2\nu}\right)^{2Q\mu k} \\
\leq\ &(2/\nu)^{2Q\mu k}\exp(-2Q\mu k/6) \\
\leq\ &(2/\nu)(\delta/16)\exp\left(-\frac{5}{2}k\right),
\end{aligned}
$$

where the second inequality holds since $\frac{\mu}{\theta_{\mathrm{avg}}(U)} \geq \nu$, and the third inequality holds since $e/4 < e^{-1/6}$. Thus, we complete the proof of (21).

Notice that the size of $\mathcal{U}$ is at most $\binom{|S|}{k}$. Together, by the union bound, the probability of premature event for a specific $\mu$ is at most $(\delta/4)/(2\nu)$. We use $\mu_{\min}$ to denote the minimum $\mu$ that is greater than $\theta_{\mathrm{avg}}(B)$. Let $\nu_{\min} = \frac{\mu_{\min}}{\theta_k(B)}$. Summing over all $\mu > \theta_{\mathrm{avg}}(B)$, we assert that the probability of premature for QE-AS is at most

$$\frac{\delta/4}{2\nu_{\min}} + \frac{\delta/4}{4\nu_{\min}} + \frac{\delta/4}{8\nu_{\min}} + \ldots \leq \frac{\delta/4}{\nu_{\min}} \leq \delta/4.$$

$\square$

The rest of the proof is almost the same as that for ME-AS in Appendix B.2, which leads us to:

**Theorem 5** *QE-AS solves the $k_{\mathrm{avg}}$-AS problem with expected cost* $O\left( \frac{n}{\epsilon^2} \frac{1}{\theta_{\mathrm{avg}}(B)} \left( 1 + \frac{\log(1/\delta)}{k} \right) \right)$.

### E.3 Lower Bound

This subsection serves as a proof for:

**Theorem 6** *For any* $\epsilon \in \left(0, \frac{1}{12}\right)$ *and* $\delta \in \left(0, \frac{1}{48}\right)$, *given any* $(\epsilon, \delta)$-*approximate algorithm, there is an instance of the $k_{\mathrm{avg}}$-AS problem on which the algorithm must entail* $\Omega\left( \frac{n}{\epsilon^2} \frac{1}{\theta_{\mathrm{avg}}(B)} \left( 1 + \frac{\log(1/\delta)}{k} \right) \right)$ *cost in expectation.*

We show that the lower bound of $k_{\mathrm{avg}}$-AS is the maximum of $\Omega\left( \frac{n}{\epsilon^2} \frac{1}{\theta_{\mathrm{avg}}(B)} \frac{\log(1/\delta)}{k} \right)$ and $\Omega\left( \frac{n}{\epsilon^2} \frac{1}{\theta_{\mathrm{avg}}(B)} \right)$.

**First Lower Bound.** We reduce 1-AS to this problem. Suppose that for 1-AS, we have a bandit with $n'$ arms. Let $I = \{a_1, \ldots, a_{n'}\}$. We associate an instance with $I$ where exactly one arm in $I$ has mean $(1 + 4\epsilon)\theta$ whereas all the other arms have mean $\theta$. Without loss of generality, we assume that $a_1$ has mean $(1 + 4\epsilon)\theta$. Recall that in Theorem 2, we showed that for any $(\epsilon, \delta)$-approximate 1-AS algorithm, distinguishing the single arm with mean $(1 + 4\epsilon)\theta$ in $I$ takes $\Omega((n' \log \frac{1}{\delta})/(\epsilon^2 \theta))$ samples. We construct a hard instance for $k_{\mathrm{avg}}$-AS based on $I$.

Let $n = kn'$. We create $n$ artificial arms as the input to algorithm $\mathcal{A}$ and divide them into $n'$ groups. Each group contains exactly $k$ arms. The $i$-th group contains the arms with indexes from $(i-1)k+1$ to $ik$. Each time when $\mathcal{A}$ attempts to sample an arm in the $i$-th group, we actually take a sample from $a_i \in I$ and passes the result to $\mathcal{A}$. If there exists an $i$ such that more than $\frac{2k}{3}$ arms returned by $\mathcal{A}$ are from group $i$, we select $a_i \in I$ as the answer for 1-AS. Otherwise, we select an arbitrary arm. Suppose that $\mathcal{A}$ can solve $k_{\mathrm{avg}}$-AS within sample complexity $Q$. Since $\epsilon < 1/12$, by definition of $k_{\mathrm{avg}}$-AS, any feasible solution must contain at least $2k/3$ arms in the first group. Thus, with probability at least $1 - \delta$, we select $a_1$ as the answer for 1-AS. Hence, $\mathcal{A}$ can solve 1-AS on the instance $I$ with $n' = n/k$ arms and $\theta = \theta_{\mathrm{avg}}(B)$, using $Q$ samples as well. Therefore, we conclude that

$$Q = \Omega\left( \frac{n'}{\epsilon^2} \frac{1}{\theta} \log \frac{1}{\delta} \right) = \Omega\left( \frac{n}{\epsilon^2} \frac{1}{\theta_{\mathrm{avg}}(B)} \frac{\log(1/\delta)}{k} \right).$$

**Second Lower Bound.** We define the *single-arm* problem as follows: with probability at least 0.6, distinguish whether a single-arm has mean $\theta$ or $(1 + 4\epsilon)\theta$.

**Lemma 17** *For any $\epsilon \in (0, \frac{1}{12})$, any algorithm solving the above single-arm problem must entail* $\Omega(\frac{1}{\epsilon^2} \frac{1}{\theta})$ *cost.*

*Proof*: Our proof is based on a sampling lower bound established in [3, Theorem 4.7]. We need the following definition:

**Definition 2** [Hellinger distance] *Suppose $X$ and $Y$ are two discrete distributions with a common support $\mathcal{S}$. The Hellinger distance between $X$ and $Y$ is defined as*

$$d_H(X, Y) = \sqrt{1 - \sum_{s \in \mathcal{S}} (\Pr[X = s])^{1/2} (\Pr[Y = s])^{1/2}}.$$

It is shown in [3] that given $0 < \delta < \frac{1}{4}$ and two distributions $X$, $Y$ on a common support with $d_H^2(X, Y) \leq \frac{1}{2}$, any algorithm distinguishing $X$ and $Y$ with failure error bounded by $\delta$ requires at least $\frac{1}{4d_H^2(X,Y)} \ln \frac{1}{\delta}$ samples. We use $U_\theta$ to denote the Bernoulli distribution with mean $\theta$ (similarly for $U_{(1+4\epsilon)\theta}$). From Definition 2, we have that

$$\begin{aligned}
& d_H^2(U_\theta, U_{(1+4\epsilon)\theta}) \\
=\ & 1 - \sqrt{\theta \cdot (1 + 4\epsilon)\theta} - \sqrt{(1-\theta)(1 - (1+4\epsilon)\theta)} \\
\leq\ & 16\epsilon^2\theta.
\end{aligned}$$

Therefore, any algorithm distinguishing two Bernoulli distribution with mean $\theta$ and $(1 + 4\epsilon)\theta$ must take at least $\Omega(\frac{1}{\epsilon^2}\frac{1}{\theta})$ samples. $\square$

The second lower bound $\Omega(\frac{n}{\epsilon^2}\frac{1}{\theta_{\text{avg}}(B)})$ is established by a reduction from the above single-arm problem. The proof is somewhat similar to the reduction in the proof of Lemma 11. First we show that for any $\epsilon \in (0, \frac{1}{12}), \delta \in (0, \frac{1}{48})$, given an $(\epsilon, \delta)$-approximate $k_{\text{avg}}$-AS algorithm $\mathcal{A}$ with expected sample complexity $Q$, we can design an algorithm $\mathcal{B}$ to solve the single-arm problem using at most $64Q/n$ samples.

Create a set of $n$ artificial arms denoted as $U = \{a_1, \dots, a_n\}$. We randomly choose a $k$-sized subset $S \subseteq U$ and an arm $a_i \in S$ as the *pivot arm*. For any arm $a_j \in S$ with $j \neq i$, we set $\theta(a_j) = (1+4\epsilon)\theta$ and for each arm $a_j \in U \backslash S$, we set $\theta(a_j) = \theta$.

Algorithm $\mathcal{B}$ solves the single-arm problem by simulating algorithm $\mathcal{A}$. When $\mathcal{A}$ samples from an arm $a_j$ with $j \neq i$, $\mathcal{B}$ simply do the same. On the other hand, when $\mathcal{A}$ samples the pivot arm $a_i$, $\mathcal{B}$ samples from arm $a$ as in the single-arm problem, and passes the result to $\mathcal{A}$. If (i) the pivot arm is sampled less than $\frac{64Q}{n}$ times and at the same time (ii) $\mathcal{A}$ returns an arm set $V$ not containing $a_i$, $\mathcal{B}$ decides that $\theta(a) = \theta$. In all other cases, $\mathcal{B}$ decides that $\theta(a) = (1 + 4\epsilon)\theta$.

Consider the case of $\theta(a) = \theta$. We use $T_a$ to denote the number of samples taken form $a$. Let $S' = S \backslash \{a_i\}$. Since $a_i$ is uniformly distributed in $U \backslash S'$ and $k \leq n/2$, we have:

$$\mathbb{E}[T_a] \leq \frac{Q}{n - k + 1} \leq \frac{2Q}{n}.$$

Therefore, $\Pr[T_a \geq \frac{64}{n}] \leq \frac{1}{32}$. If $T_a < \frac{64}{n}$, $\mathcal{B}$ errs if and only if $a_i \in V$ holds. Notice that in this case, any feasible solution must contain at least $2k/3 - 1$ arms with mean $(1+4\epsilon)\theta$. Therefore with probability at least $1 - \delta$, $\mathcal{A}$ returns at most $\frac{k}{3} + 1$ arms with mean $\theta$. Thus,

$$\Pr[a_i \in V] \leq \delta + (1 - \delta)\frac{k/3 + 1}{n - k + 1} \leq 0.35.$$

Thus, with probability at least $1 - 0.35 - 1/32 \geq 0.62$, $\mathcal{B}$ outputs the correct answer.

Next, consider the case where $\theta(a) = (1 + 4\epsilon)\theta$. In this case, any feasible solution must contain at least $2k/3$ arms with mean $(1 + 4\epsilon)\theta$. $\mathcal{B}$ errs only if $T_a < 64Q/n$ and $a_i \notin V$. Since $\mathcal{A}$ is an $(\epsilon, \delta)$-approximate $k_{\text{avg}}$-AS algorithm, we have that $\Pr[a_i \in V] \geq \frac{2}{3}(1 - \delta) \geq 0.65$. Therefore, by the lower bound for the single-arm problem, we have that $Q = \Omega\left(\frac{n}{\epsilon^2}\frac{1}{\theta}\right) = \Omega\left(\frac{n}{\epsilon^2}\frac{1}{\theta_{\text{avg}}(B)}\right)$.

# F  Analysis of $k_{\text{avg}}$-MOST CONNECTED VERTEX

Our $k_{\text{avg}}$-AS algorithm, combined with the reduction described in Section 1.1, already settles $k_{\text{avg}}$-MCV with the sample complexity given in Table 1. Next, we prove the following lower bound.

**Theorem 7** *For any $\epsilon \in \left(0, \frac{1}{12}\right)$ and $\delta \in \left(0, \frac{1}{48}\right)$, the following statements are true about any $k_{\text{avg}}$-MCV algorithm:*

- *when* $\deg_{\mathsf{avg}}(B) \geq \Omega\left(\frac{1}{\epsilon^2}\log\frac{n}{\delta}\right)$, *there is an instance on which the algorithm must probe*

$$\Omega\left(\frac{n}{\epsilon^2}\frac{m}{\deg_{\mathsf{avg}}(B)}\left(1 + \frac{\log(1/\delta)}{k}\right)\right)$$

*edges in expectation.*

- *when* $\deg_k(B) < O(\frac{1}{\epsilon})$, *there is an instance on which the algorithm must probe* $\Omega(nm)$ *edges in expectation.*

**Case** $\deg_{\mathsf{avg}}(B) \geq \Omega((1/\epsilon^2)\log\frac{n}{\delta})$. We first show a lower bound of $\Omega((nm\log\frac{1}{\delta})/(\epsilon^2 k \deg_{\mathsf{avg}}(B)))$, using the hard 1-AS instance constructed in the proof of Theorem 6. Recall that the instance has a set $I$ of $n'$ arms $\{a_1,\ldots,a_{n'}\}$. Construct from $I$ a hidden bipartite graph $G = (B,W,E)$ where $|B| = kn' = n$ and $|W| = m \geq (24\log\frac{n}{\delta})/(\epsilon^2\theta)$. Divide $B$ into $n'$ groups, each having $k$ vertices. For each black vertex in the $i$-th group, we independently make each edge incident on it solid with probability $\theta(a_i)$.

Let $\mathcal{A}$ be an algorithm solving the $k_{\mathsf{avg}}$-MCV problem with sample complexity $Q$. We feed $G$ as an input to $\mathcal{A}$. If more than $k/2$ vertices returned by $\mathcal{A}$ are from the group $i$, we return $a_i$ as the answer for 1-AS. Otherwise, we return an arbitrary arm.

Set $t = m\theta$. The same argument in the proof of Theorem 4 shows that if $t \geq \frac{24}{\epsilon^2}\log\frac{n}{\delta}$, with probability at least $1 - \delta$, the degree of each black vertex in the first group is no less than $(1 + 3\epsilon)t$, and the degree of any other black vertex is no more than $(1 + \epsilon/2)t$. Thus, with probability at least $1 - 2\delta$, $\mathcal{A}$ returns a vertex set containing more than $k/2$ vertices from the first group. We thus have actually obtained an $(\epsilon, 2\delta)$-approximate algorithm for 1-AS on instance $I$. However, we already know that any $(\epsilon, \delta)$-algorithm solving instance $I$ requires $\Omega((n'\log\frac{1}{\delta})/(\epsilon^2\theta))$ samples. As $n = kn'$, it follows that $Q = \Omega((nm\log\frac{1}{\delta})/(\epsilon^2 k \deg_{\mathsf{avg}}(B)))$.

We then prove a lower bound of $\Omega(nm/(\epsilon^2\deg_{\mathsf{avg}}(B)))$. Using the hard instance constructed in the proof of the second lower bound of Theorem 6, we construct a random hidden bipartite graph $G = (B,W,E)$ with $|B| = n$ and $|W| = m \geq (24\log\frac{n}{\delta})/(\epsilon^2\theta)$ in the same way as in proving Theorem 4. We can show that with probability $1 - \frac{\delta}{n}$, for each vertex $b$, $\deg(b)/m$ is within in $(1 \pm \epsilon)$ factor of the mean of the corresponding arm. Following essentially the same argument as Theorem 6, we can show that, if an algorithm $\mathcal{A}$ can $(\epsilon, \delta)$-approximate $k_{\mathsf{avg}}$-MCV using $Q$ samples, then we can solve the single-arm problem using $O(\frac{Q}{n})$ samples. Hence we can get $Q = \Omega(\frac{n}{\epsilon^2}\frac{1}{\theta}) = \Omega(\frac{n}{\epsilon^2}\frac{m}{\deg_{\mathsf{avg}}(B)})$.

**Case** $\deg_{\mathsf{avg}}(B) < O(1/\epsilon)$. The proof is almost the same as the second lower bound of Theorem 6. The only difference is to reduce the problem to the single vertex problem.

# G  Additional Experiments

We provide the additional experiment results in this section. First, in Appendix G.1 we give a performance evaluation of $k$-MCV vs. $\delta$ as the supplementary of Section 6. Then, we provide the experimental result for $k$-AS problem in Appendix G.2. Finally, we evaluate QE-AS for both $k_{\mathsf{avg}}$-MCV and $k$-AS problems in Appendix G.3 and discuss the experimental results in Appendix G.4

(a) Power law with $\bar{deg} = 50$   (b) Power law with $\bar{deg} = 3000$   (c) 2-hop

Figure 2: Performance comparison for $k$-MCV vs. $\delta$

(a) Power law with $\bar{deg} = 50$     (b) Power law with $\bar{deg} = 3000$     (c) 2-hop

Figure 3: Performance comparison for $k$-AS vs. $\epsilon$

(a) Power law with $\bar{deg} = 50$     (b) Power law with $\bar{deg} = 3000$     (c) 2-hop

Figure 4: Performance comparison for $k$-AS vs. $\delta$

(a) Power law with $\bar{deg} = 50$     (b) Power law with $\bar{deg} = 3000$     (c) 2-hop

Figure 5: Performance comparison for $k_{\mathsf{avg}}$-MCV vs. $\epsilon$

(a) Power law with $\bar{deg} = 50$     (b) Power law with $\bar{deg} = 3000$     (c) 2-hop

Figure 6: Performance comparison for $k_{\mathsf{avg}}$-AS vs. $\epsilon$

## G.1    Evaluation for $k$-MCV vs. $\delta$

We fix the parameter $k = 20, \epsilon = 0.05$ and enumerate $\delta$ from 0.05 to 0.5. Again, we label the actual error $\epsilon_a$ whenever the algorithm does not achieve the theoretical guarantees. The results are shown in Figure 2.

## G.2    Evaluation for $k$-AS

We use the same method as in Section 6 to build two bandit instances where the means of the arms follow the power law distribution. Furthermore, recall that in Section 6, we build a bipartite graph

for the 2-hop relationships with $|B| = n$ and $|W| = m$. Utilizing such bipartite graph, we build a bandit instance with $n$ arms where the mean of the $k$-th arm equals $\deg_k(B)/m$.

Again, we fix $k = 20$ and enumerate $\epsilon, \delta$ separately. We label the actual error $\epsilon_a$ whenever the algorithm does not achieve the theoretical guarantees. The results are shown in Figure 3 and Figure 4.

### G.3 Evaluation for $k_{\mathsf{avg}}$-MCV and $k_{\mathsf{avg}}$-AS

The performance of QE-AS is stable when $\delta$ varies. Therefore, we only evaluate the performance of QE-AS vs $\epsilon$ for both $k_{\mathsf{avg}}$-MCV and $k_{\mathsf{avg}}$-AS here, as shown in Figure 5 and Figure 6.

### G.4 Discussion

The experimental results match our theoretical analysis. First we show a performance comparison for $k$-MCV vs $\delta$. Again, ME-AS outperforms AMCV in both sample cost and the actual error occurred when $\delta$ varies. Similarly, we draw the same conclusion for these two algorithms on $k$-AS. Moreover, as we claimed in Section 1, it costs less samples for $k$-MCV than $k$-AS for a specific $(\epsilon, \delta)$ since we do not need to probe the same edge twice. Finally, the experimental results also show $k$-AS ($k$-MCV) usually require more samples than $k_{\mathsf{avg}}$-AS ($k_{\mathsf{avg}}$-MCV).