[Reviews · NeurIPS 2015]

Submitted by Assigned_Reviewer_1

Pros considers more useful criteria than prior works more useful optimality: multiplicative error stronger optimality even in additive error: 1-to-1 optimality (or, element-wise threshold) rather than many-to-one optimality (group threshold). improved lower bounds. proposes algorithms that matches the lower bound

Cons Does not work with unbounded rewards with sub-Gaussian since the \mu search cannot start from 1 in that case. Most existing top-k selection algorithms works well with sub-Gaussian rewards as well.

\delta \in (0, 1/48) is quite a strong assumption involving ME typically leads to a poor performance in practice, although it is order-optimal. For hidden bipartite, the bounds are vacuous unless the number of graphs is extremely large.

Questions line 143: why abs-err weaker than rel-err? Is it because simply the additive "error" resulting from \eps-rel-err definition smaller than \eps? Perhaps explain it with a few sentences. Algorithm 1: why try \mu=1 ? the loop termination condition will always fail.
Summary: Overall, good theory paper that sets up lower bounds for the proposed criteria. However, the algorithms that match the lower bounds are based on repeatedly calling median elimination algorithm that notoriously requires large number of samples.

Submitted by Assigned_Reviewer_2

Summary - - - - - - -

This paper studies the problem of finding the top k (out of n) arms (the k arms with the highest mean) in multi-armed bandits fixed confidence setting. More specifically, the goal is to derive an algorithm that with the smallest number of pulls returns a set of k arms such that with high probability (w.p. \geq 1-\delta)

1) for each 1 \leq i \leq k, the mean of the i'th arm in the set is larger than (1-\epsilon) times the mean of the i'th arm (overall).

2) the average mean of the arms returned by the algorithm is larger than (1-\epsilon) times the average mean of the top k arms.

The above settings are different and more general than those have been considered in the multi-armed bandits fixed confidence pure exploration literature (e.g., the work by Kalyanakrishnan et al., 2012 and the one by Gabillon et al, 2012), in the sense that the algorithms in those papers guarantee that every arm they return is (\epsilon,k)-optimal (the mean of every arm returned is not smaller than the mean of the k'th best arm minus \epsilon, with high probability).

For each of the above two problems, the authors present an algorithm and prove an upper-bound and a matching lower-bound for it. The authors motivate these two settings with the problem of crowdsourcing. The authors also show how their algorithm for the first problem should be changed to solve the same problem with additive instead of multiplicative guarantee (the mean of the i'th arm plus \epsilon instead of (1-\epsilon) times the mean of the i'th arm).

Then the authors focus on the problem of exploring hidden bipartite graphs and state two problems, similar to those stated above. These two graph problems can be easily reduced to the two problems above, and thus, the algorithms for those problems can be easily used here. The only contribution in this part of the paper is the derivation of a lower-bound for each of these two graph problems.

Comments - - - - - - - -

- The paper is well-written and is easy to follow.

- To the best of my knowledge, the formulation considered in this paper is novel. The objective functions studied here (in Problems 1 and 2), although close, are different than those studied in the literature of pure exploration multi-armed bandits with fixed confidence. They are different in two senses: 1) the mean of each returned arm should be close to the mean of a specific top k arm (instead of being close to the mean of the k'th arm), and 2) the guarantee is multiplicative instead of additive ((1-\epsilon) times the mean of the i'th arm instead of the mean of the i'th arm plus \epsilon). The first difference could play an important role in the case where there exists many arms whose mean is in [\theta_k(B) - \epsilon, \theta_k(B)]. In this case, the existing (\epsilon,k)-optimal algorithms may return k arms, none of them among the top k, while this is not the case for the algorithm proposed in this paper. The authors have a discussion (on Pages 2 and 3) on the significance of multiplicative guarantee and mention its importance when the optimal solution is very small, which is apparently the case in many hidden graph problems. This is a good reason to consider multiplicative guarantee in addition to additive. They also discuss how we can define \epsilon'=\epsilon\theta_k(B), and use algorithms for additive guarantee to have multiplicative guarantee.

- In addition to the objective function, there is a very important difference between the top-k algorithms in the literature (e.g., the work by Kalyanakrishnan et al., 2012 and the one by Gabillon et al, 2012) and the one here, which is in the bound when the complexity appears in their bounds instead of 1 / \epsilon^2*\theta_k(B). This means that the bounds of the existing methods can be much tighter than the one for the algorithm in this paper. This is the main reason that those algorithms are more complex than the racing type algorithms presented in this paper. My main criticism about this work is the lack of discussion about this issue. As I mentioned above I admit that the formulation considered in the paper is novel, but the authors have considered the simplest approach (the racing type algorithms) and of course their bound is of the form 1 / \epsilon^2\theta_k(B). I think it would be possible to make the complexity appear in the bound for the problems considered in this paper, of course (probably) not by the type of algorithms considered here. It would be great if the authors make this difference clear and discuss the difficulties of having algorithms with the complexity in their bounds for the problems studied in this paper. Why didn't the authors derived algorithms of that form?

- Why the proposed algorithms use the empirical mean based on the uniform allocation at that specific iteration instead of the empirical mean based on all the available samples from that arm? Is this just because of simplifying the proof and theory or has other reasons? Please clarify.

Minor Comments - - - - - - - - - - - -

- Line 102: "instead an additive" should be "instead of an additive".

- Why the guess for the value of the mean of the k'th arm (\mu) starts at \mu=1 in ME-AS and QE-AS algorithms? This seems to be useless since \hat{\theta}^{US}(a_k) will never be greater than 2*\mu=2. I guess it should start at 1/2.
Summary: (+) The formulation considered in the paper is novel (to the best of my knowledge).

(+) The authors address several problems: 1) top-k arm selection with additive and multiplicative guarantees, 2) top-k-average arm selection, for both they propose an algorithm with upper and lower bounds, and 3) similar problems with graphs for which the only contribution is the lower-bound.

(+) They motivate the proposed settings (I do not consider this part of the paper strong, but not much worse than the other papers in this genre) and support their results with simple experiments.

(-) The lack of discussion about the difference in the bound with the existing results (see my comment on this). The fact that several existing results are based on more complex algorithms than the racing type algorithms considered in this paper, mainly because they want tighter bounds with the complexity appearing in the bound, instead of 1 / \epsilon^2\theta_k(B).

Submitted by Assigned_Reviewer_3

Motivated by several applications, the authors study the pure exploration phase of multi-armed bandits, and the closely related setting of hidden bipartite graphs. The aim being to identify the top k arms while minimizing the number of arm pulls.

More precisely, in the multi-armed bandit setting they establish matching upper and lower bounds for identifying with high probability the top-k (up to a multiplicative constant) arms by reward and the set of k-arms with (approximately) highest average reward. For the hidden bipartite problem (finite bipartite graph with hidden edge set), they study the problem of finding the top-k (again up to a multiplicative constant) degree vertices on one side and the (approximately) highest average degree. For the hidden bipartite graph problem they establish new upper and lower bounds on the sample complexity, these bounds leave a small gap in their applicability and come close to completing the picture for these problems.

The main algorithm at a high level is intuitive and can be described as a two-stage sample-sort-discard approach, but the details are tricky involving careful parameter tuning in applying the measure concentration results. The authors do a good job of keeping the reader abreast of what is going on, even if the details are relegated to the appendix. The upper bounds for the graph versions are provided by reduction to the bandit setting. The lower bounds are established using two techniques, when the k-th largest degree is large they can provide a reduction from the bandit setting to the graph version. When the k-th largest degree is small they provide a nice proof by utilizing sampling complexity to distinguishing two very similar distributions. As alluded above there is a gap of applicability of the two lower bounds but this is great progress.

The results are interesting and important, the paper itself is very well written and I recommend acceptance.

Minor comments: - I found the notation \theta_avg(V) and deg_avg(V) confusing since it suppresses the quantity k present in their definition. Perhaps use \theta_k-avg(V) or something like that. - The comments on page 4, lines 199-201 "Currently, no lower bound results are known..." is confusing since you provide new lower bounds for this case. Perhaps re-write as "Prior to this work, no lower bound results were known..." - I couldn't easily find the definition of Top-k_add (defined in lines ~211...213) while reading about it later in the paper. Making this more readily visible would be appreciated.
Summary: The authors make a number of contributions to the study of the pure exploration phase of multi-armed bandits problem and the related problem of finding high degree vertices in hidden bipartite graphs. They provide new more efficient algorithms for the problem of (approximately) identifying the best subset of k-arms or the set with highest average reward. They also prove sharper lower bounds and for the bipartite graph versions provide the first such bounds.

Submitted by Assigned_Reviewer_4

The paper considers the problem of top-k arm selection allowing for small approximation error (\epsilon). The error is defined in a relative manner unlike to the EXPLORE-k problem where the error is defined in an absolute manner. Moreover, the top-k arms selection problem is defined in a more strict way, that is, the means of the selected arms must match to best k arms respectively in a decreasing order up to some relative error. This problem is called k-AS. In addition, an aggregated goal is also considered where the average of the means of the selected arm cannot be much worse than the average of the means of all arms (again up to some relative error). A related, non-stochastic version of these problems are also studied which are called k-MCV and k_{avg}-MCV.

Lower and upper bounds for the expected sample complexity are provided for the considered problems. The proposed algorithm is a nice interplay of the median elimination strategy by Even-dar et.al. 2006 and the one proposed by Sheng, Tao, and Li, 2012. The median elimination first roughly filter out the very bad arms and then based on a quality check (called US) the algorithm decides either to terminate or to continue sampling. This algorithm can be directly applied to all problems that are considered in the paper in an elegant way.

The experiments on twitter data verify that the proposed algorithm outperforms the existing one proposed by Sheng, Tao, and Li, 2012.

I liked the paper. I have only some minor comments which might ease the understanding of the paper (see below). In my opinion, this paper is too long for a conference. I have to admit that I did not check the whole appendix which consists of 20 pages!! I read through only Appendix B which contains the analysis of the proposed algorithm for k-AS problem.

I think at least the basic idea of the proof should be described in the paper.

Major comments:

1) Regarding the problem definition given in line 065, at this point it is not clear why \epsilon and \delta must fall into the given intervals. Does this really belong to the problem definition? Similarly in line 090. And moreover it is not clear what is the role of \delta in the case of k-MCV since there is no randomness here. This property is called ``history-awareness property'' in line 128. If randomised algorithm is considered, then this might be not the part of the problem definition.

2.) Regarding the paragraph starting at line 145: frist, it is not clear how \Theta_k ( B ) = o(1) is meant. \Theta_k ( B ) is a constant determined by the problem instance. So how its limiting behaviour is meant here? I guess it is meant as k goes to |B|. Right? Then \Theta_k ( B ) = o(1) implies that there must be an arm whose mean is equal to zero. I think this notation should be used here.

3.) It would be worth to point out more explicitly in the text that the sample complexity is meant in expectation ( I've found this in the caption of Table 1). But first I believed that the sample complexity bounds are PAC bounds like the ones considered by Even-dar et al 2006.

Minor comment:

line 424: Victor Gabillon et al. -> Gabillon et al.
Summary: The paper is well-written and does make a significant enough contribution to warrant publication.

Author Feedback
Author rebuttal: *Reviewer 1*
Q1: (1) lack of discussion about the bounds in Kalyanakrishnan et al., 2012 and Gabillon et al, 2012 and the comparison with the bounds in this work
(2) why didn't the authors derived algorithms with gap dependent bound

A1: Kalyanakrishnan et al., 2012 and Gabillon et al, 2012 considered the Explorer-k setting and provided the algorithms using ``lower and upper confidence bound" (LUCB). Their sample complexity for Explorer-k problem is O(H^{\epsilon/2} log (H^{\epsilon / 2} / \delta)). In worst case scenarios, this can be as high as O(n/(\epsilon^2)\log(n/\dleta)), which is a logn/logk factor worse than our bound. In the scenarios where the gaps of a lot of arms are large, their bound may be better than ours. Thanks for this comment, and we will make this comparison more transparent in the next version.
For the second question, it is unclear right now how to adapt the LUCB framework to solve our AS problem. Basically, LUCB maintains a lower and upper confidence bound for each arm, and pulls (one of) the two arms which are most difficult to distinguish, i.e., the arm with the smallest lower confidence bound among the empirically k largest arms and the arm with the largest upper confidence bound among the other (n - k) arms. The algorithm terminates when these two arms are well-separated. However, in our setting, since our goal is to ensure an individual error guarantee for all top-k arms (stronger than Explorer-k), it is unclear whether only sampling the most difficult-to-distinguish arms would be enough. Nevertheless, this does not rule out the possibility of an efficient algorithm with a nice gap dependent bound. We leave it as an intriguing direction for future work.

Q2: use the empirical mean based on the uniform allocation at that specific iteration

A2: Our main reason to use empirical means based on the uniform allocation at that specific iteration is to simplify the theoretical analysis. We believe it is possible to use all the available samples. However, the current theoretical analysis needs to be modified accordingly (and it seems to us the modification would not be trivial).

Q3: why \mu start at 1
A3: Indeed it is better to start from \mu = 1/2 although it does not affect the complexity. We will fix it in the next version.

*Reviewer 2*
Thank you very much for your encouraging review! We will incorporate your comments in our paper.

*Review 3*
Q1: definition of \epsilon and \delta
A1: In most applications, we only care about the case where the error \eps and failure probability \delta are small. Note that if \eps is close to 1, the arm we selected can be arbitrarily bad since (1 - \epsilon) is nearly zero. Moreover, the concentration inequalities we used in our analysis require \epsilon to be in a certain range (lines 613, 636 etc). The same reason applies to \delta.

Q2: randomized algorithm in k-MCV
A2: Thanks for the insightful comment. Indeed, in k-MCV, there is no randomness in the given instance. Hence, it only makes sense to consider randomized algorithms. The reason is that for any deterministic algorithm, the adversary can force the algorithm to always probe \Omega(mn) edges, which makes the problem uninteresting. That is why our problem formulation allows approximate answers. This is a good point worth clarifying in the next version.

Q3: how \Theta_k ( B ) = o(1) is meant
A3: We meant that \Theta_k ( B ) is very small. Clearly this can be stated in a better way which we will do in the next section.

Q4: Explicitly point out the complexity is meant in expectation:
A4: Thank you for your suggestion. We will fix it.

*Review 7*
Q1: sub-Gaussian
A1: Yes, currently the algorithm does not work with sub-Gaussian. We will point it out in the next version, and study it in our future work.

Q2: abs-err weaker than rel-err
A2: Thanks for the comment. You are right. Since for each arm \theta \in (0, 1), the relative error is always smaller than the absolute error. Especially, \theta can be extremely small in some applications. For such cases, the relative guarantee is much stronger. We will make this point clearer in the next version.

Q3: \delta in (0, 1/48)
A3: The concentration inequalities we used in our analysis require a certain range of \delta. However, by more carefully tuning the parameters, we can relax the restriction of 1/48.

Q4: median elimination requires a large number of samples
A4: We agree that in practice the elimination algorithm indeed requires many samples. However, currently we do not see a way reduce the sample size (note: UCB/LUCB type algorithms may not work in our one-to-one optimality setting; see our response to Reviewer 1 Q1). We leave the development of practical algorithms with optimal gap-dependent bounds as important future work.

Q5: why \mu start from 1
A5: Indeed it is better to start from \mu = 1/2 (even though it does not affect the complexity). We will fix it in our paper.